# Combining viral genomics and clinical data to assess risk factors for severe COVID-19 (mortality, ICU admission, or intubation) amongst hospital patients in a large acute UK NHS hospital Trust

Max Foxley-Marrable[1], Leon D'Cruz[2], Paul Meredith[2], Sharon Glaysher[2], Angela H. Beckett[3,4], Salman Goudarzi[5], Christopher Fearn[5], Kate F. Cook[5], Katie F. Loveson[5], Hannah Dent[5], Hannah Paul[5], Scott Elliott[2], Sarah Wyllie[2], Allyson Lloyd[2], Kelly Bicknell[2], Sally Lumley[2], James McNicholas[2], David Prytherch[6], The COVID-19 Genomics UK (COG-UK) consortium[¶], Andrew Lundgren[1], Or Graur[1], Anoop J. Chauhan[2], Samuel C. Robson[3,4,5]*

1 Institute of Cosmology and Gravitation, University of Portsmouth, Portsmouth, Hampshire, United Kingdom, 2 Portsmouth Hospitals University NHS Trust, Portsmouth, Hampshire, United Kingdom, 3 School of Biological Science, University of Portsmouth, Portsmouth, Hampshire, United Kingdom, 4 Centre for Enzyme Innovation, University of Portsmouth, Portsmouth, Hampshire, United Kingdom, 5 School of Pharmacy and Biomedical Science, University of Portsmouth, Portsmouth, Hampshire, United Kingdom, 6 Centre for Healthcare Modelling and Informatics, University of Portsmouth, Portsmouth, Hampshire, United Kingdom

¶ Full list of consortium names and affiliations is in the Supporting information (S1 File). www.cogconsortium.uk
* samuel.robson@port.ac.uk

**Data Availability Statement:** The resulting consensus SARS-CoV-2 genomes and human-

## Abstract

Throughout the COVID-19 pandemic, valuable datasets have been collected on the effects of the virus SARS-CoV-2. In this study, we combined whole genome sequencing data with clinical data (including clinical outcomes, demographics, comorbidity, treatment information) for 929 patient cases seen at a large UK hospital Trust between March 2020 and May 2021. We identified associations between acute physiological status and three measures of disease severity; admission to the intensive care unit (ICU), requirement for intubation, and mortality. Whilst the maximum National Early Warning Score (NEWS2) was moderately associated with severe COVID-19 ($A = 0.48$), the admission NEWS2 was only weakly associated ($A = 0.17$), suggesting it is ineffective as an early predictor of severity. Patient outcome was weakly associated with myriad factors linked to acute physiological status and human genetics, including age, sex and pre-existing conditions. Overall, we found no significant links between viral genomics and severe outcomes, but saw evidence that variant sub-type may impact relative risk for certain sub-populations. Specific mutations of SARS-CoV-2 appear to have little impact on overall severity risk in these data, suggesting that emerging SARS-CoV-2 variants do not result in more severe patient outcomes. However, our results show that determining a causal relationship between mutations and severe COVID-19 in the viral genome is challenging. Whilst improved understanding of the evolution of SARS-CoV-

filtered sequencing data for COG-UK samples are routinely deposited in the European Nucleotide Archive (ENA) at EMBL-EBI under accession PRJEB37886 (https://www.ncbi.nlm.nih.gov/bioproject/?term=PRJEB37886). In addition, high-quality consensus genome files with coverage greater than 90% are routinely deposited to the Global Initiative for Sharing of All Influenza Data (GISAID) database (https://gisaid.org/). Aggregated clinical data are provided within the manuscript and its Supporting Information files. Raw clinical data cannot be shared publicly because of risks to patient confidentiality. Data are available from the Portsmouth Hospitals University NHS Trust Institutional Data Access / Ethics Committee (contact via research.office@porthosp.nhs.uk) and may be made available for researchers who meet the criteria for access to confidential data.

**Funding:** This work was primarily funded by the COVID-19 Genomics UK (COG-UK) consortium (https://www.cogconsortium.uk/), under their Internal Principal Investigator Research Funding Scheme. COG-UK is supported by funding from the Medical Research Council (MRC; https://www.ukri.org/councils/mrc/) part of UK Research & Innovation (UKRI; https://www.ukri.org/), the National Institute of Health Research (NIHR; https://www.nihr.ac.uk/) [grant code: MC_PC_19027], and Genome Research Limited, operating as the Wellcome Sanger Institute (https://www.sanger.ac.uk/). The authors acknowledge the use of data generated through the COVID-19 Genomics Programme funded by the Department of Health and Social Care (DHSC; https://www.gov.uk/government/organisations/department-of-health-and-social-care). The views expressed are those of the author and not necessarily those of the Department of Health and Social Care or PHE or UKHSA. MFM, AL, and OG were also supported by a UKRI Science & Technology Facilities Council (STFC) Impact Accelerator Account awarded to the Institute of Cosmology and Gravitation (ICG) at the University of Portsmouth. Additional funding for the project came from the University of Portsmouth Faculty of Science and Health (https://www.port.ac.uk/about-us/structure-and-governance/organisational-structure/our-academic-structure/faculty-of-science-and-health), and the Wessex Academic Health Sciences Centre (AHSC; https://wessexahsn.org.uk/). In addition, SCR and AHB are funded by Research England's Expanding Excellence in England (E3) Fund. The primary funders had no role in study design, data collection and analysis, decision to publish, or manuscript preparation.

2 has been achieved through genomics, few studies on how these evolutionary changes impact on clinical outcomes have been seen due to complexities associated with data linkage. By combining viral genomics with patient records in a large acute UK hospital, this study represents a significant resource for understanding risk factors associated with COVID-19 severity. However, further understanding will likely arise from studies of the role of host genetics on disease progression.

## Introduction

Coronavirus disease 2019 (COVID-19), caused by the Severe Acute Respiratory Syndrome Coronavirus-2 (SARS-CoV-2) pathogen [1], has resulted in arguably the most significant global health crisis in recent history. SARS-CoV-2 was first identified in Wuhan, China in winter 2019 [2] and quickly spread across the globe, being declared a pandemic by the World Health Organisation (WHO) a few months later in March 2020 [3]. At the time of writing, COVID-19 has resulted in over 625 million infections and 6.57 million deaths worldwide [4]. As well as this significant death toll, many survivors have suffered life-altering complications as a result of contracting the disease [5]. The COVID-19 pandemic has also resulted in significant social and economic disruption, including the biggest global recession since the Great Depression [6]. Additionally, healthcare services such as the UK National Health Service (NHS) have been significantly impacted by COVID-19, resulting in staff shortages, long wait times for ambulances [7] and a significant backlog for patients needing elective care [8].

The factors that influence severe cases of COVID-19 are not yet fully understood but have been clearly linked primarily to older age groups (predominantly the over 65s), primarily due to a higher proportion of comorbidities [9]. However, younger patients still experience severe outcomes from COVID-19, albeit more rarely. Severe outcomes include requirements for intubation and mechanical ventilation, admission to intensive care units (ICU), and death. Factors currently associated with an increased risk of severe outcomes include smoking, having a pre-existing condition such as obesity, asthma, cardiovascular disease, and diabetes, or socio-economic factors [9–11]. Children especially are far less likely to become seriously ill from COVID-19 [12,13].

Large-scale SARS-CoV-2 sequencing programs throughout the pandemic have allowed researchers to explore the role of viral genomics and different variants of the virus. However, whilst genomic epidemiology has been used in a number of studies to understand viral transmission in settings such as hospitals [14–17], long-term care facilities [18–22], and army barracks [23,24], well-powered studies of patient outcomes with high numbers of cases currently remain limited [25–31]. One of the largest studies looking at large-scale effects of viral genomics on patient outcomes are papers from the Hospital Onset COVID-19 Infections (HOCI) study in the UK [25,26,28]. In one such study, Stirrup et al. (2021) identified a higher hazard ratio of mortality for female patients with the Alpha variant compared to other variants when compared to male patients [26]. More recently, Webster *et al.* [30] identified lower or equivalent risk of severe outcomes for the BA.2 Omicron variant compared to BA.1.

Here we combined data from two resources developed over the course of the pandemic in the city of Portsmouth in the UK; whole-genome sequencing (WGS) of SARS-CoV-2 samples from COVID-19 positive samples collected through the COVID-19 Genomics UK (COG-UK) Consortium by researchers at the University of Portsmouth (UoP), and patient-specific information (e.g. demographics, COVID-19 status, illness severity scores, comorbidities, treatments

**Competing interests:** Scott Elliot and Salman Goudarzi currently work for QIAGEN, UK, however were employees of Portsmouth Hospitals University NHS Trust and the University of Portsmouth respectively when the work described in this manuscript was conducted. QIAGEN had no role in the study design, data collection and analysis, decision to publish, or manuscript preparation, and this does not alter our adherence to PLOS ONE policies on sharing data and materials. The remaining authors declare that no competing interests exist.

and outcomes) for all hospital admissions collected by the Portsmouth Academic Consortium For Investigating COVID-19 (PACIFIC-19) team at Portsmouth Hospitals University NHS Trust (PHU). PHU saw a steep rise in COVID-19 cases over the winter of 2020, with 3,272 new hospital cases between September and February, and a peak of 539 positive inpatients, representing a national outlier for infections compared to the average peak of 219 in the South East (https://coronavirus.data.gov.uk/).

The aim of this study was to combine the COG-UK dataset and PACIFIC-19 Clinical Outcomes Research Group (CORG) database to develop a data resource linking clinical disease severity, therapeutic interventions, comorbidities and demographics to SARS-COV-2 genomic lineage data. One such metric, the National Early Warning Score 2 (NEWS2), provides a simple metric for identifying acutely ill patients and those requiring transfer to ICU [32,33]. It is calculated based on 6 physiological parameters recorded at the bedside (respiration rate, oxygen saturation, systolic blood pressure, pulse rate, level of consciousness or new-onset confusion, temperature), each assigned a score of 0–3 by the healthcare team, with a score greater than 7 suggesting a high-risk patient requiring emergency assessment by the critical care team. These data cover COVID-19 infections in the area between March 2020 and May 2021, including the major UK wave of COVID-19 over winter 2020, and were used to explore factors influencing clinical severity of COVID-19 and identify specific mutations or constellations of mutations associated with severe COVID-19. In particular, this time period covers the introduction of the first variant of concern (VOC) Alpha, known also by the Pangolin (https://cov-lineages.org/) lineage name B.1.1.7, allowing us to address whether the emergence of this lineage impacted on the clinical severity of COVID-19.

As global restrictions continue to flex in response to ongoing changes in case-loads, and we learn to live with the SARS-CoV-2 virus as new VOCs develop, it is increasingly important to look back at what we have learned to fully understand the factors associated with poor outcomes from COVID-19. There is significant motivation to further expand our knowledge of potential risk factors for severe COVID-19, especially where such factors may allow medical staff to predict a severe outcome of COVID-19 for early intervention. This study thus provides a significant resource for understanding the role that a variety of clinical factors and viral genomics play in determining patient outcomes.

## Materials and methods

### Study sites

PHU is one of England's largest acute hospital trusts, serving the major coastal port city of Portsmouth and surrounding areas on the South Coast of the UK. The primary site for this study was Queen Alexandra Hospital (QAH), a research hospital within PHU with an 800-bed capacity treating >500,000 patients per year.

### Laboratory diagnosis

Quantitative polymerase chain reaction (qPCR) COVID-19 tests for hospital staff, patients, and members of the local community within Portsmouth and surrounding areas were carried out at QAH. Samples were collected from participants using nasopharyngeal swabs and stored and transported in Sigma-Virocult 1 mL Viral Transport Media (VTM) (Medical Wire & Equipment, Corsham, UK).

Multiple clinically validated testing methods were used over the period of the study, following manufacturer's directions. These approaches include using the Panther system with the Aptima SARS-CoV-2 assay (Hologic, Marlborough, USA). This method involves automated RNA extraction and transcription-mediated amplification, providing a qualitative result to

confirm the presence or absence of SARS-CoV-2 by amplifying two conserved regions of the SARS-CoV-2 ORF1ab gene, comparing the fluorescence signal to an internal control.

Additional testing was performed using the Anatolia Geneworks SARS-CoV-2 PCR v2 kit, which has 2 SARS-CoV-2 targets: ORF1ab and E gene alongside an internal control. VTM sample extraction was performed on the QIAsymphony SP/AS extraction system (Qiagen, Hilden, Germany) off-board lysis protocol (PATHOGEN, COMPLEX 200_OBL_V4_DSP) using the QIAsymphony DSP Virus/Pathogen Midi or Mini Kit and reverse transcription (RT) real-time qPCR amplification was performed on the LightCycler 480 II (Roche, Basel, Switzerland).

Additional rapid testing was conducted using the Xpert® Xpress SARS-CoV-2 assay on the GeneXpert (Cepheid, California, USA), a cartridge-based system for rapid detection, extraction and amplification using real-time RT-qPCR to detect 2 targets for SAR-COV-2 in the N2 and E gene regions, alongside internal controls.

## Sampling

All samples, including patients, healthcare workers (HCWs) and community cases tested for COVID-19 at PHU, were made available for viral extraction and whole genome sequencing. Samples from PHU were sequenced alongside samples from a wide range of NHS Trusts across the South Coast of the UK by the University of Portsmouth as part of the COG-UK consortium [34]. Where samples could not be sequenced due to limits in capacity, the COG-UK surveillance sampling strategy was applied to ensure that cases represented a random representation of currently circulating variants. Briefly, samples were selected either due to targeted sequencing priorities, such as HCWs for the SARS-CoV-2 Immunity & Reinfection EvaluatioN (SIREN) study (https://snapsurvey.phe.org.uk/siren/), or were selected randomly from available samples each day up to local capacity.

## Whole genome sequencing

Sequencing was conducted following the ARTIC nCoV-2019 sequencing protocol V.3 (LoCost) [35]. RNA was reverse transcribed and then amplified with amplicon PCR using the ARTIC nCoV-2019 V3 primer panel (Integrated DNA Technologies, Iowa, USA). This primer panel tiles the SARS-CoV-2 genome with 98 pairs of primers, each producing an amplicon of ~500 bp. Odd-numbered primers were pooled separately from even-numbered primers to prevent over-amplification of overlapping amplicon regions.

Nuclease-free water (NFW) was used as a negative control on each sequencing run to assess contamination in the amplification stage. A synthetic SARS-CoV-2 RNA control (Twist Bioscience, San Francisco, CA, USA) was also added to each run as a positive control. To confirm sample quality and assess likely failures or contamination issues, positive and negative controls, along with representative samples from each run, were quantified using the Qubit DNA Assay Kit in a Qubit 2.0 Fluorometer (Life Technologies, California, USA).

The LSK-109 Ligation Sequencing Kit and EXP-NBD196 Native Barcoding Expansion 96 Kit from Oxford Nanopore Technologies (ONT, Oxford, UK) were used to generate libraries for Nanopore sequencing. Libraries were sequenced on R9.4.1 flow cells on a GridION X5 platform (ONT, Oxford, UK) for 24–36 hours (depending on library sample number) to achieve a final coverage of ~100,000 reads per sample. Raw reads were demultiplexed by the MINKnow software on the GridION using Guppy v3.2.10.

Sequencing data were processed using the ARTIC field bioinformatics toolkit v1.2.1 (https://github.com/artic-network/artic-ncov2019). Real-time sequencing performance was monitored using RAMPART (v1.0.6) [36]. Reads were mapped to the SARS-CoV-2 reference genome (Wuhan-Hu-1, GenBank, MN908947.3) using MiniMap2 (v2.17-r941) [37].

Nucleotide variation from the reference sequence was identified using Nanopolish (v0.13.2; https://github.com/jts/nanopolish). SARS-CoV-2 variant type was assigned using Pangolin (https://github.com/cov-lineages/pangolin) with PANGOLearn version 2021-10-18.

## Sample exclusion

If genome sequencing failed (e.g., as a result of the negative control showing evidence of PCR contamination), samples were repeated from scratch. If sufficient RNA was not available, samples were excluded from the study. Samples from PHU were also excluded if the participant involved indicated their retrospective desire to opt out from the study.

For the outcome analysis, further exclusions were also applied to the combined dataset. Samples where the sequence data covered less than 50% of the genome were excluded due to poor resolution of viral variant subclasses. Samples were also excluded for individuals aged less than 16 years old, individuals that were not admitted to the main hospital (e.g., residents of long-term care facilities), and individuals who had not yet completed their hospital stay. In cases where multiple samples were taken from a single individual, the sample with the highest genome coverage was taken forward for further analysis. This is summarised in S1 Fig.

## Clinical outcome data

The PACIFIC-19 team at PHU holds a database of patient-specific information (e.g. demographics, COVID-19 status, illness severity scores, treatments and outcomes) for all hospital admissions, including COVID-19 positive patients, between January 2018 and May 2021. The PACIFIC-19 CORG database contains data collated from the Local Laboratory Information Systems (LIMS) using COGNOS for interrogation to identify all positive samples, and manually from the APEX Pathology LIMS. These data were linked to SARS-CoV-2 genome sequence data using the COG-UK sequencing codes and locally assigned sample source IDs.

## Clinical data analysis

To maximise the number of near-complete entries usable for our analyses, we dropped data columns where 15% or more of the entries contained missing data. Imputation of missing values was not used to avoid significantly biassing the results.

Three main measures of severity as a result of COVID-19 infection were used in this analysis; patient death within 30 days of diagnosis, patient admission to ICU or intubation of the patient. In addition, we took a general measure of case severity based on the occurrence of at least one of these three outcomes.

For pair-wise associations between categorical variables, the association strength was calculated using Cramer's V score $V$ (with bias correction) [38], based on the $\chi^2$ statistic, with statistical significance calculated using the p-value from a $\chi^2$ test [39]. For pair-wise associations between continuous variables, the correlation coefficient $\rho$ and p-value from a Spearman's Rank test were used to determine the association strength and statistical significance respectively. For pair-wise associations between categorical and continuous variables, the association strength was determined using the Correlation Ratio $\eta^2$ [40].

To ensure no bias as a result of non-normally distributed data, the continuous variable was ranked prior to calculation. Statistical significance was determined using the p-value from a Kruskal–Wallis H test. In each case, the association strength score $A$ was assumed to be negligible if $|A| < 0.1$, weak if $0.1 \geq |A| > 0.3$, moderate if $0.3 \geq |A| > 0.5$, and strong if $|A| \geq 0.5$. Associations were determined to be statistically significant when $p < 0.05$.

## Machine learning for the identification of mutations associated with disease severity

Mutation information from sequencing experiments was numerically encoded as follows: 1 = wild-type, 2 = substitution, 3 = insertion, 4 = deletion. These data were linked to clinical data as input for machine learning models to further explore the role of viral mutations of SARS-CoV-2 on severity of disease in COVID-19. We screened nine machine learning models and one deep-learning neural network method to rank and identify mutations with a possible role in determining patient outcomes. Training of models and calculation of accuracy metrics were determined from 6-fold stratified cross-validation screening using Python V3.8.8 with TensorFlow V2. Data were proportioned into an 80:20 train-test split.

A binary-outcome variable for severity was defined based on mortality having occurred following escalation to the ICU. To address the imbalance in these data, with 3.2-fold fewer cases of mortality than survival, Synthetic Minority Oversampling (SMOTE) techniques were implemented. Hyperparameters were tuned for optimal performance using a Grid-search method while implementing 6-fold cross-validation. To get an overall view of the metrics incorporating both classes, precision, recall and F1 statistics were calculated for cases in the test set with outcome = 0 or outcome = 1 separately, with the macro-average scores calculated based on the mean of the two.

The best accuracy combined with minimal loss scores were obtained using the multi-layer perceptron artificial neural network (MLP-ANN), using the sequential API within TensorFlow. The input layer to the MLP-ANN introduces linear weighted input variables to the neurons in the hidden-layers. Dropout regularization was employed to offset the overfitting dilemma typically encountered in machine-learning models [41]. This approximates training of a large number of neural networks with different architectures in parallel, where a number of layers are randomly ignored or dropped out. Model accuracy and loss scores began to plateau by 4,000 epochs, so were run to 10,000 epochs to maximise the accuracy (S2 Fig).

## Ethics statement

This work has been approved by the Health Research Authority (HRA) and Health and Care Research Wales (HCRW) following a favourable opinion from the North West–Haydock Research Ethics Committee on 24[th] April 2020 (Ref: 20/NW/0217). Participants were offered the opportunity to opt out of having their anonymised data used in this study retrospectively. This work is part of the Sequencing and Tracking of Phylogeny (STOP COVID-19) study, which was posted to ClinicalTrials.gov (Ref: NCT04359849) on 24[th] April 2020. This work also forms part of the wider COVID-19 Genomics UK (COG-UK) Consortium surveillance study, which was approved by the Public Health England Research Ethics Governance Group and granted ethical approval by the PHE Research Ethics and Governance Group (REGG) on 8[th] April 2020, (PHE R&D ref: R&D NR0195). The PACIFIC-19 Clinical Outcomes Research Group (CORG) database was approved by the HRA Research Ethics Committee in April 2021 (Ref: 21/SC/0080), with a study extension provided to allow access to the data for the STOP COVID-19 project (IRAS 282394).

## Results

### Patient demographics

The primary dataset used in this analysis combines viral genomics with clinical metadata for PHU. Following filtering of cases (see Materials and methods) and merging of the data sets, combined data for 929 individual patients were used for downstream analyses (S1 Fig). A

breakdown of these data based on some of the key demographics and clinical factors can be seen in Table 1. Of these 929 cases, 360 (38.8%) showed severe outcomes (ICU admission, intubation or death within 30 days of diagnosis), with 569 (61.2%) showing non-severe outcomes. Looking at the severe outcomes in more detail, 295 (31.8%) patients died, 111 (11.9%) patients were admitted to ICU, and 93 (10.0%) required intubation in ICU. Of those patients on ICU, 46 (41.4%) also died, suggesting that the majority of fatalities (249; 84.4%) occurred outside of ICU, with 70 (23.7%) occurring outside of the hospital. However, the majority of these deaths (181; 61.4%) occurred in patients aged 80 or above, with only 5 admitted to ICU. In general, patients suffering severe outcomes were older, with a median age of 76 (IQR [63,85]), with 52.8% of cases between 70 and 90 years old. The split between male and female cases was relatively even, with 426 (45.8%) female compared with 503 (54.1%) male cases. The majority of all cases were of white ethnic background (701; 75.5%), 192 (20.7%) cases were of unstated or unknown ethnic origin and the remaining 36 (3.8%) cases were comprised of non-white ethnic minority groups.

At the time of COVID-19 diagnosis, almost half of all patients were inpatients (450 cases, 48.4%), with a large proportion being identified through the Emergency Department (ED; 360 cases, 38.8%). A smaller proportion of cases were identified in Critical Care (CC; 62 cases, 6.7%) and the Acute Medical Units (AMU; 33 cases, 3.6%). The majority of patients suffered from at least one of the comorbidities (801 cases, 86.2%) explored in this dataset; diabetes, hypertension, renal disease, malignancy (cancer), heart disease, asthma, or chronic obstructive pulmonary disease (COPD). Hypertension and heart disease were the most common, with 485 (52.2%) and 490 (52.7%) cases respectively, whilst asthma and cancer were rarer, with 101 (10.9%) and 108 (11.6%) cases respectively.

## Associations with disease severity

To understand the factors that most affect disease severity (defined by either admission to ICU, receiving invasive mechanical intubation, or death within 30 days of diagnosis), pairwise statistical association analyses were performed for all experimental variables using either Cramer's V, Spearman's Rank or the Correlation Ratio, depending on the data types (see Materials and methods). S3 Fig shows the pairwise association score between all variables in the data set, and Table 2 shows those with a statistically significant ($p < = 0.05$) and non-negligible ($A \geq 0.1$) association with COVID-19 severity. A description of these data points is shown in S1 Table. In general, these data show that clinical variables show mild association with outcomes, and demonstrate a lack of individual strong indicators in our dataset that could potentially predict a severe case of COVID-19.

The maximum NEWS2 score showed the highest association with severe cases of COVID-19 ($A = 0.48$, p = 5.10e-46; Fig 1A), indicating a moderate but statistically significant association. In particular, this metric was more strongly associated with death ($A = 0.43$, p = 3.49e-37) than with ICU admission ($A = 0.21$, p = 2.30e-09) or intubation ($A = 0.21$, p = 2.52e-09). The maximum NEWS2 score shows median scores of 9 (high-risk; IQR [6,10]) for severe cases and 5 (medium-risk; IQR [3,7]) for non-severe cases. The NEWS2 score assigned to a patient on admission is also associated with severity, albeit much more weakly overall ($A = 0.17$, p = 1.11e-08; Fig 1B). Interestingly, whilst associated with ICU admission ($A = 0.24$, p = 1.15e-16) and intubation ($A = 0.24$, p = 5.61e-17), the admission NEWS2 score was not associated with patient death alone. The admission NEWS2 score shows median scores of 4 (IQR [1,8]) for severe cases and 3 (IQR [1,5]) for non-severe cases with a large amount of overlap between the distributions, indicating that the NEWS2 score at admission may not be a strong initial predictor of severe COVID-19.

**Table 1. Patient data summary.**

| | All Cases | Non-Severe | Severe | Fatal | ICU | Intubation |
|---|---|---|---|---|---|---|
| **All Cases** | 929 | 569 (61.2%) | 360 (38.8%) | 295 (31.8%) | 111 (11.9%) | 93 (10.0%) |
| **Admission Age** | 76 [63, 85] | 74 [60, 85] | 80 [68, 86] | 82 [73, 88] | 63 [54, 71] | 64 [56, 71] |
| 0–59 | 192 (20.7%) | 142 (15.3%) | 50 (5.4%) | 16 (1.7%) | 44 (4.7%) | 33 (3.6%) |
| 60–69 | 133 (14.3%) | 83 (8.9%) | 50 (5.4%) | 29 (3.1%) | 37 (4.0%) | 34 (3.7%) |
| 70–79 | 205 (22.1%) | 126 (13.6%) | 79 (8.5%) | 69 (7.4%) | 25 (2.7%) | 21 (2.3%) |
| 80–89 | 285 (30.7%) | 156 (16.8%) | 129 (13.9%) | 129 (13.9%) | 5 (0.5%) | 5 (0.5%) |
| 90–99 | 109 (11.7%) | 60 (6.5%) | 49 (5.3%) | 49 (5.3%) | - | - |
| 100+ | 5 (0.5%) | 2 (0.2%) | 3 (0.3%) | 3 (0.3%) | - | - |
| **Length of Stay (days)** | 13 [6, 23] | 11 [5, 20] | 16 [8, 26] | 14 [6, 22] | 25 [16, 54] | 25 [16, 54] |
| **Time to Discharge or Death from Diagnosis (hours)** | 201 [98, 383] | 164 [87, 344] | 245 [123, 502] | 202 [110, 325] | 546 [351, 1033] | 584 [355, 1073] |
| **Admission NEWS2 Score** | 3 [1, 6] | 3 [1, 5] | 4 [1, 8] | 4 [1, 7] | 7 [4, 10] | 8 [5, 9] |
| **Maximum NEWS2 Score** | 6 [4, 9] | 5 [3, 7] | 9 [6, 10] | 9 [6, 11] | 8 [7, 10] | 8 [7, 10] |
| **Sex** | - | - | - | - | - | - |
| Male | 503 (54.1%) | 276 (29.7%) | 227 (24.4%) | 180 (19.4%) | 79 (8.5%) | 64 (6.9%) |
| Female | 426 (45.8%) | 293 (31.5%) | 133 (14.3%) | 115 (12.4%) | 32 (3.4%) | 29 (3.1%) |
| **Ethnic Origin** | - | - | - | - | - | - |
| Asian | 16 (1.7%) | 8 (0.9%) | 8 (0.9%) | 6 (0.6%) | 7 (0.8%) | 7 (0.8%) |
| Black | 6 (0.6%) | 3 (0.3%) | 3 (0.3%) | 1 (0.1%) | 3 (0.3%) | 1 (0.1%) |
| Mixed | 8 (0.9%) | 4 (0.4%) | 4 (0.4%) | 1 (0.1%) | 4 (0.4%) | 2 (0.2%) |
| Other | 6 (0.6%) | 6 (0.6%) | - | - | - | - |
| Unknown | 192 (20.7%) | 122 (13.1%) | 70 (7.5%) | 59 (6.4%) | 18 (1.9%) | 19 (2.0%) |
| White | 701 (75.5%) | 426 (45.9%) | 275 (29.6%) | 228 (24.5%) | 79 (8.5%) | 64 (6.9%) |
| **Lineage** | - | - | - | - | - | - |
| Alpha | 404 (43.5%) | 255 (27.4%) | 149 (16.0%) | 116 (12.5%) | 57 (6.1%) | 51 (5.5%) |
| Non-Alpha | 525 (56.5%) | 314 (33.8%) | 211 (22.7%) | 179 (19.3%) | 54 (5.8%) | 42 (4.5%) |
| **Patient Type** | - | - | - | - | - | - |
| Inpatients | 450 (48.4%) | 290 (31.2%) | 160 (17.2%) | 147 (15.8%) | 18 (1.9%) | 12 (1.3%) |
| Emergency Department | 360 (38.8%) | 238 (25.6%) | 122 (13.1%) | 108 (11.6%) | 29 (3.1%) | 29 (3.1%) |
| Critical Care | 62 (6.7%) | 3 (0.3%) | 59 (6.4%) | 23 (2.5%) | 59 (6.4%) | 49 (5.3%) |
| Acute Medical Unit | 33 (3.6%) | 21 (2.3%) | 12 (1.3%) | 11 (1.2%) | 3 (0.3%) | 2 (0.2%) |
| Outpatients | 15 (1.6%) | 10 (1.1%) | 5 (0.5%) | 4 (0.4%) | 2 (0.2%) | 1 (0.1%) |
| Healthcare Workers | 4 (0.4%) | 4 (0.4%) | 0 (0%) | - | - | - |
| Community Cases | 3 (0.3%) | 2 (0.2%) | 1 (0.1%) | 1 (0.1%) | - | - |
| External PHU Hospital Patients | 1 (0.1%) | 1 (0.1%) | 0 (0%) | - | - | - |
| Long-Term Care Facility Residents | 1 (0.1%) | 0 (0%) | 1 (0.1%) | 1 (0.1%) | - | - |
| **Diabetes** | - | - | - | - | - | - |
| 0 (No) | 636 (68.5%) | 403 (43.4%) | 233 (25.1%) | 195 (21.0%) | 61 (6.6%) | 52 (5.6%) |
| 1 (Yes) | 293 (31.5%) | 166 (17.9%) | 127 (13.7%) | 100 (10.8%) | 50 (5.4%) | 41 (4.4%) |
| **Hypertension** | - | - | - | - | - | - |
| 0 (No) | 444 (47.8%) | 293 (31.5%) | 151 (16.3%) | 123 (13.2%) | 45 (4.8%) | 37 (4.0%) |
| 1 (Yes) | 485 (52.2%) | 276 (29.7%) | 209 (22.5%) | 172 (18.5%) | 66 (7.1%) | 56 (6.0%) |
| **Renal Disease** | - | - | - | - | - | - |
| 0 (No) | 583 (62.8%) | 401 (43.2%) | 182 (19.6%) | 151 (16.3%) | 49 (5.3%) | 42 (4.5%) |
| 1 (Yes) | 346 (37.2%) | 168 (18.1%) | 178 (19.2%) | 144 (15.5%) | 62 (6.7%) | 51 (5.5%) |
| **Malignancy** | - | - | - | - | - | - |
| 0 (No) | 821 (88.4%) | 513 (55.2%) | 308 (33.2%) | 246 (26.5%) | 104 (11.2%) | 87 (9.4%) |
| 1 (Yes) | 108 (11.6%) | 56 (6.0%) | 52 (5.6%) | 49 (5.3%) | 7 (0.8%) | 6 (0.6%) |

*(Continued)*

**Table 1.** (Continued)

| | All Cases | Non-Severe | Severe | Fatal | ICU | Intubation |
|---|---|---|---|---|---|---|
| **Heart Disease** | - | - | - | - | - | - |
| 0 (No) | 439 (47.3%) | 298 (32.1%) | 141 (15.2%) | 107 (11.5%) | 53 (5.7%) | 44 (4.7%) |
| 1 (Yes) | 490 (52.7%) | 271 (29.2%) | 219 (23.6%) | 198 (20.2%) | 58 (6.2%) | 49 (5.3%) |
| **Asthma** | - | - | - | - | - | - |
| 0 (No) | 828 (89.1%) | 504 (54.3%) | 324 (34.9%) | 267 (28.s%) | 98 (10.5%) | 80 (8.6%) |
| 1 (Yes) | 101 (10.9%) | 65 (7.0%) | 36 (3.9%) | 28 (3.0%) | 13 (1.4%) | 13 (1.4%) |
| **COPD** | - | - | - | - | . | . |
| 0 (No) | 760 (81.8%) | 477 (51.3%) | 283 (30.5%) | 225 (24.2%) | 96 (10.3%) | 80 (8.6%) |
| 1 (Yes) | 169 (18.2%) | 92 (9.9%) | 77 (8.3%) | 70 (7.5%) | 15 (1.6%) | 13 (1.4%) |
| **Number of Pre-Existing Conditions** | - | - | - | - | - | . |
| 0 | 128 (13.8%) | 102 (11.0%) | 26 (2.8%) | 19 (2.0%) | 10 (1.1%) | 6 (0.6%) |
| 1 | 220 (23.7%) | 145 (15.6%) | 75 (8.1%) | 59 (6.4%) | 24 (2.6%) | 22 (2.4%) |
| 2 | 261 (28.1%) | 161 (17.3%) | 100 (10.8%) | 88 (9.5%) | 23 (2.5%) | 22 (2.4%) |
| 3 | 212 (22.8%) | 112 (12.1%) | 100 (10.8%) | 79 (8.5%) | 32 (3.4%) | 24 (2.6%) |
| 4 | 95 (10.2%) | 46 (5.0%) | 49 (5.3%) | 41 (4.4%) | 20 (2.2%) | 17 (1.8%) |
| 5 | 13 (1.4%) | 3 (0.3%) | 10 (1.1%) | 9 (1.0%) | 2 (0.2%) | 2 (0.2%) |

Summary table breaking down key variables from the joint dataset for all, non-severe and severe cases, respectively. Data presented is either the number of cases and percentage of all cases for categorical variables, or the median with the interquartile range for continuous variables.

Moderate to weak associations were also seen with the ward location category ($A = 0.28$, p = 4.50e-16), specific ward ($A = 0.21$, p = 7.88e-06) and the location where the qPCR swab was originally collected ($A = 0.16$, p = 9.74e-03). All three are associated with ICU admission and intubation, but not with patient death as an outcome (Table 2). All three were also associated with admission NEWS2 score (ward location category $A = 0.44$, p = 1.59e-37; specific ward $A = 0.60$, p = 3.15e-49; qPCR swab location $A = 0.56$, p = 8.61e-38) and maximum NEWS2 score (ward location category $A = 016$, p = 1.27e-03; specific ward $A = 0.33$, p = 1.56e-06; qPCR swab location $A = 0.31$, p = 6.08e-03), as well as whether the patient was admitted to ICU (ward location category $A = 0.66$, p = 2.39e-89; specific ward $A = 0.49$, p = 1.63e-43; qPCR swab location $A = 0.43$, p = 9.61e-24). Together, these indicate that patients who suffered from a severe case of COVID-19 were typically acutely unwell in general and therefore more likely to be transferred to ICU.

Pre-existing comorbidities also appeared to play a role in susceptibility for severe COVID-19, with a weak association seen for the number of pre-existing conditions a patient might have ($A = 0.20$, p = 6.81e-09), as well as a weak association to those who have any pre-existing conditions ($A = 0.13$, p = 2.05e-05; Fig 2A). These links were seen with the death outcome, but not with ICU admission nor intubation (Table 2). Specifically, those with renal disease ($A = 0.19$, p = 6.32e-09; Fig 2B) or heart disease ($A = 0.12$, p = 3.10e-04; Fig 2C) showed weak but statistically significant associations with COVID-19 severity, in particular death. These links therefore result in increased odds of having a severe case of COVID-19 (pre-existing condition OR = 2.81, 95% CI [1.79, 4.42]; renal disease OR = 2.33, 95% CI [1.77, 3.07]; heart disease OR = 1.71, 95% CI [1.31, 2.24]).

Demographics such as age ($A = 0.15$, p = 1.00e-04; Fig 1C) and sex ($A = 0.13$, p = 6.05e-05; Fig 2A) of the patient also show statistically significant, albeit weak effects on COVID-19 severity. These data show a median age of 80 (IQR [68,86]) in severe cases compared to 74 (IQR

**Table 2. Association between clinical variables and disease severity.**

| Feature | Association Strength Metric | A (Combined) | p (Combined) | A (Fatal) | p (Fatal) | A (ICU) | p (ICU) | A (Intubated) | p (Intubated) |
|---|---|---|---|---|---|---|---|---|---|
| Maximum NEWS2 Score | Correlation Ratio | 0.48 | 5.10E-46 | 0.43 | 3.49E-37 | 0.21 | 2.30E-09 | 0.21 | 2.52E-09 |
| Location Category | Cramer's V | 0.28 | 4.50E-16 | - | - | 0.66 | 2.39E-89 | 0.59 | 1.77E-71 |
| Ward | Cramer's V | 0.21 | 7.88E-06 | - | - | 0.49 | 1.63E-43 | 0.43 | 3.31E-31 |
| Number of Pre-Existing Conditions | Correlation Ratio | 0.20 | 6.81E-09 | 0.18 | 9.10E-08 | - | - | - | - |
| Renal Disease Indicator | Cramer's V | 0.19 | 6.32E-09 | 0.15 | 3.30E-06 | 0.12 | 7.40E-05 | - | - |
| Admission NEWS2 Score | Correlation Ratio | 0.17 | 1.11E-08 | - | - | 0.24 | 1.15E-16 | 0.24 | 5.61E-17 |
| Length of Stay | Correlation Ratio | 0.17 | 1.82E-07 | - | - | 0.30 | 3.51E-19 | 0.28 | 5.26E-16 |
| Swab Location | Cramer's V | 0.16 | 9.74E-03 | - | - | 0.43 | 9.61E-24 | 0.32 | 3.52E-12 |
| Time to Discharge/Death from Diagnosis | Correlation Ratio | 0.15 | 6.14E-07 | - | - | 0.36 | 5.32E-28 | 0.34 | 1.12E-24 |
| Admission Age | Correlation Ratio | 0.15 | 1.00E-04 | 0.30 | 4.62E-18 | 0.29 | 1.15E-18 | 0.25 | 4.55E-14 |
| Pre-Existing Condition Indicator | Cramer's V | 0.13 | 2.05E-05 | 0.13 | 4.78E-05 | - | - | - | - |
| Sex | Cramer's V | 0.13 | 6.05E-05 | - | - | 0.10 | 5.00E-04 | - | - |
| Heart Disease Indicator | Cramer's V | 0.12 | 3.10E-04 | 0.14 | 2.13E-05 | - | - | - | - |
| Malignancy Indicator | Cramer's V | - | - | 0.11 | 4.29E-03 | - | - | - | - |
| Admission Specialty | Cramer's V | - | - | - | - | 0.32 | 6.01E-15 | 0.32 | 2.17E-16 |
| Ethnic Origin | Cramer's V | - | - | - | - | 0.22 | 1.12E-07 | 0.22 | 4.14E-04 |
| Lineage | Cramer's V | - | - | - | - | 0.18 | 3.76E-03 | - | - |
| Diabetes Indicator | Cramer's V | - | - | - | - | 0.11 | 3.93E-03 | - | - |

Table of variables with statistically significant links with patient severity, showing the association strength A (based on Cramer's V for categorical-categorical relationships and Correlation Ratio for categorical-continuous relationships) and p-value p of the relationship. Associations with $A < 0.1$ or $p < 0.05$ are not shown.

[60,85]) in non-severe cases, and that male patients showed a higher ratio of severe to non-severe cases when compared to female patients (OR = 1.81, 95% CI [1.38, 2.37], p = 6.05e-05; Fig 2D).

In addition, the length of stay (A = 0.17, p = 1.82e-07; Fig 1D) and time to discharge or death (A = 0.15, p = 6.14e-07) also appear to be statistically significant (albeit weak) factors, with a longer median stay of 16 days (IQR [8,26]) seen in severe cases compared to 11 days (IQR [5,20]) in non-severe cases. In particular, we see a long tail for long stays for severe cases, as a result of patients who contracted severe, but non-fatal, COVID-19 and required a significant amount of recovery time. Interestingly, both metrics were associated with ICU admission and intubation, but not with patient death (Table 2).

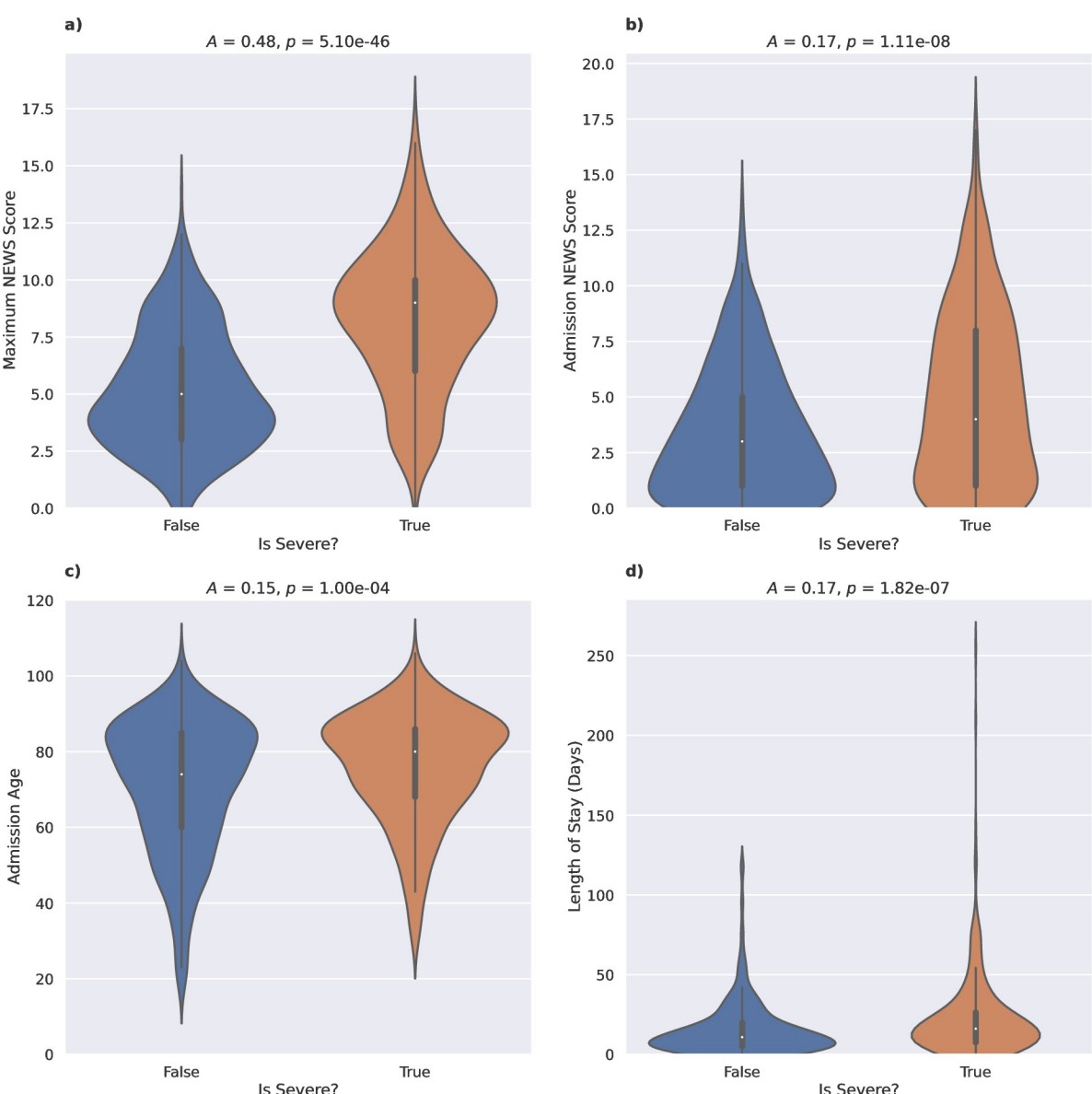

**Fig 1. Association of disease severity with continuous variables.** Violin plots comparing the distribution of continuous variables with statistically significant relationships with disease severity between severe cases (death, ICU admission or intubation) and non-severe cases. Variables shown are a) the maximum NEWS2 score, b) the admission NEWS2 score, c) the age at admission and d) the length of stay (days). Association strength *A* (based on the Correlation Ratio) and p-values *p* are shown above each panel.

## The effect of the Alpha variant on clinical severity of COVID-19

The Alpha variant (B.1.1.7) was imported to PHU during the second wave of COVID-19 cases in the UK, in particular rising in prevalence during the winter period in December 2020. Over the period between September 2020 and June 2021, 1,404 cases were sequenced from PHU, of which 970 (523 patient, 447 HCWs) proved to be of the Alpha lineage. Across the dataset used in this study, spanning cases in PHU from March 2020 until May 2021, 43.5% of COVID-19 cases were cases of the Alpha lineage (Table 1). In contrast, the next most common variant, Pangolin lineage B.1.1, comprised only 6% of cases, making Alpha the most common single variant identified by a significant margin. Across the whole dataset, we found no statistically

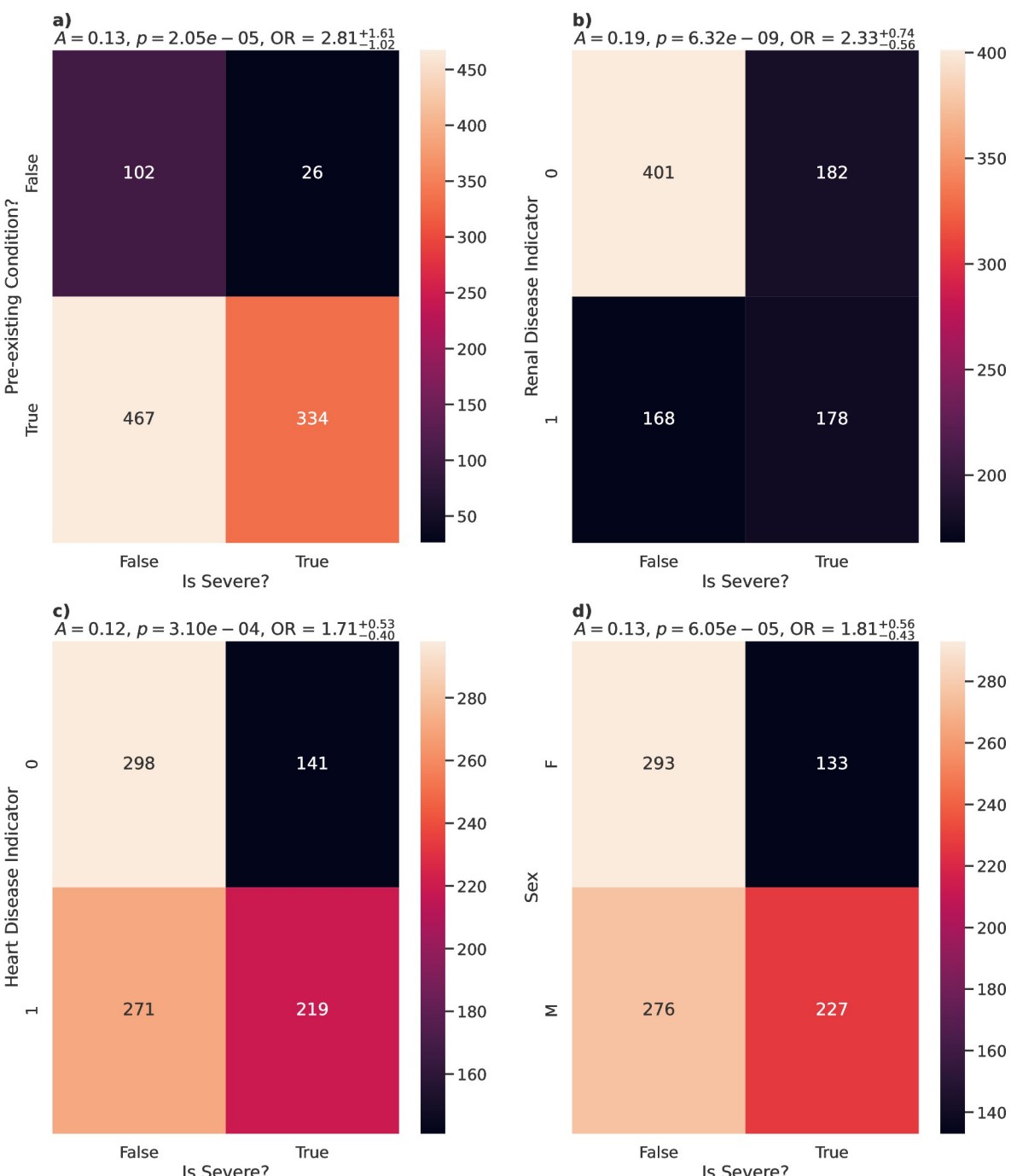

**Fig 2. Association of disease severity with discrete variables.** Heatmaps comparing the counts between severe cases (death, ICU admission or intubation) and non-severe cases for a selection of categorical features with statistically significant relationships with disease severity. Variables shown are a) the presence of existing conditions, b) whether the patient suffers from renal disease, c) whether the patient suffers from heart disease and d) sex at birth. Association strength $A$ (based on Cramer's V), p-values $p$, and odds ratio between the classes are shown above each panel.

significant link between lineage and case severity ($A = 0.08$, p = 0.356), and no statistically significant links between COVID-19 severity and whether the case was Alpha lineage or not ($A = 0.00$, p = 0.435).

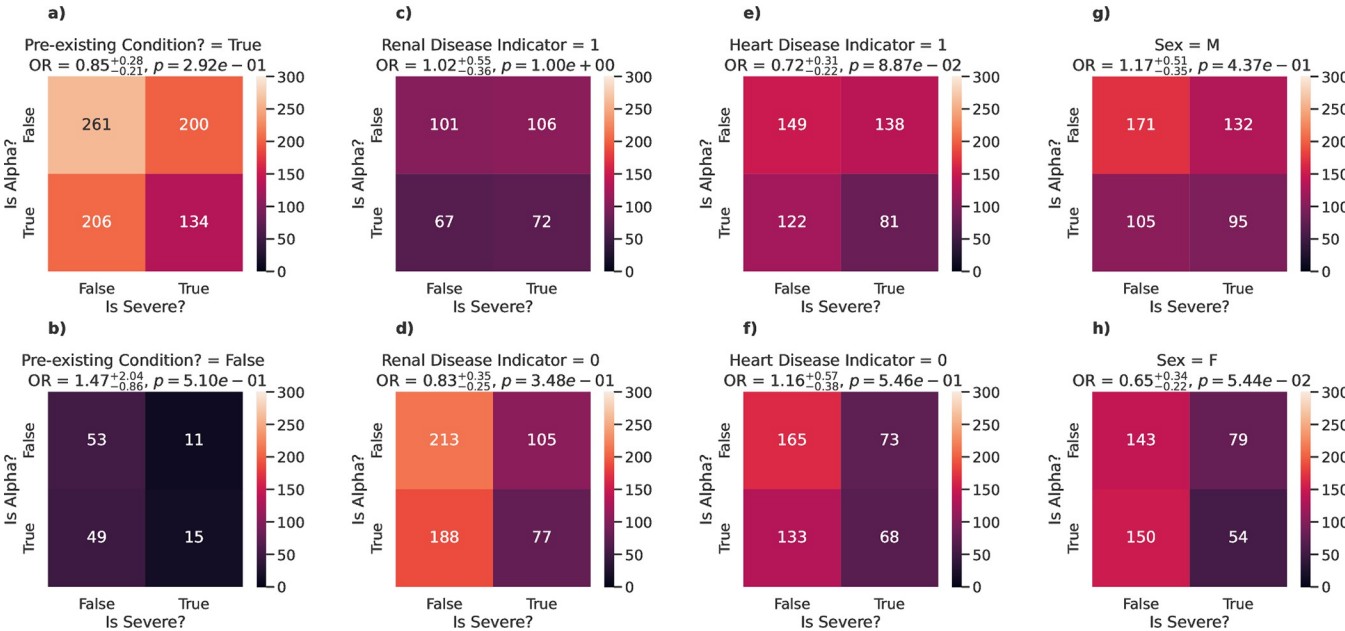

**Fig 3. Effect of the Alpha variant on disease severity.** Heatmaps exploring the changes in relative risk of severe outcome (death, ICU admission or intubation) for subpopulations within the data when comparing Alpha cases to non-Alpha cases. The subpopulations shown are based on those identified as being associated with disease severity. They are (in column order): Whether the patient has a pre-existing condition or not (a-b), whether the patient has renal disease or not (c-d), whether the patient has heart disease or not (e-f), and sex at birth (g-h). The odds ratio and p-values $p$ between the classes are shown above each panel.

To further understand the effect of the Alpha lineage on disease severity, we looked at associations with disease severity for Alpha and non-Alpha cases separately (Fig 3). Fig 3A–3F explore how the Alpha variant may have impacted severity outcomes for those with pre-existing conditions, and specifically renal and heart disease, which were identified as being associated with disease severity (Table 2). We see that the Alpha variant had no statistically significant impact on patients with (OR = 0.85, 95% CI [0.64, 1.13], p = 0.292; Fig 3A) or without (OR = 1.47, 95% CI [0.61, 3.51], p = 0.510; Fig 3B) a pre-existing condition; with (OR = 1.02, 95% CI [0.66, 1.57], p = 1.00; Fig 3C) or without (OR = 0.83, 95% CI [0.58, 1.18], p = 0.348; Fig 3D) renal disease; or with (OR = 0.72, 95% CI [0.50, 1.03], p = 0.089; Fig 3E) or without (OR = 1.16, 95% CI [0.78, 1.73], p = 0.546; Fig 3H) heart disease. Finally, we see that whilst the Alpha variant had no statistically significant impact on male patients (OR = 1.17, 95% CI [0.82, 1.68], p = 0.437; Fig 3G), female patients showed a mild but non-significant decrease in the ratio of severe cases with the Alpha variant (OR = 0.65, 95% CI [0.43, 0.99], p = 0.054; Fig 3H).

Further breaking down the relationship of the Alpha variant with severity for male and female patients, the Alpha variant showed no statistically significant impact on mortality for male patients (OR = 0.94, 95% CI [0.65, 1.37], p = 0.839; Fig 4A) nor for female patients (OR = 0.65, 95% CI [0.42, 1.00], p = 0.061; Fig 4B), although a moderate decrease in odds was seen. Interestingly, the Alpha variant did have an impact on ICU admission (OR = 2.03, 95% CI [1.25, 3.30], p = 5.50e-03; Fig 4C) and whether intubation was required (OR = 2.32, 95% CI [1.36, 3.95], p = 2.51e-03; Fig 4E) for male patients, with almost twice the odds when compared to non-Alpha variants in both cases. However, no statistically significant impact on ICU admission (OR = 0.84, 95% CI [0.41, 1.73], p = 0.762; Fig 4D) nor whether intubation was required (OR = 1.02, 95% CI [0.48, 2.17], p = 1.00; Fig 4F) was seen with female patients.

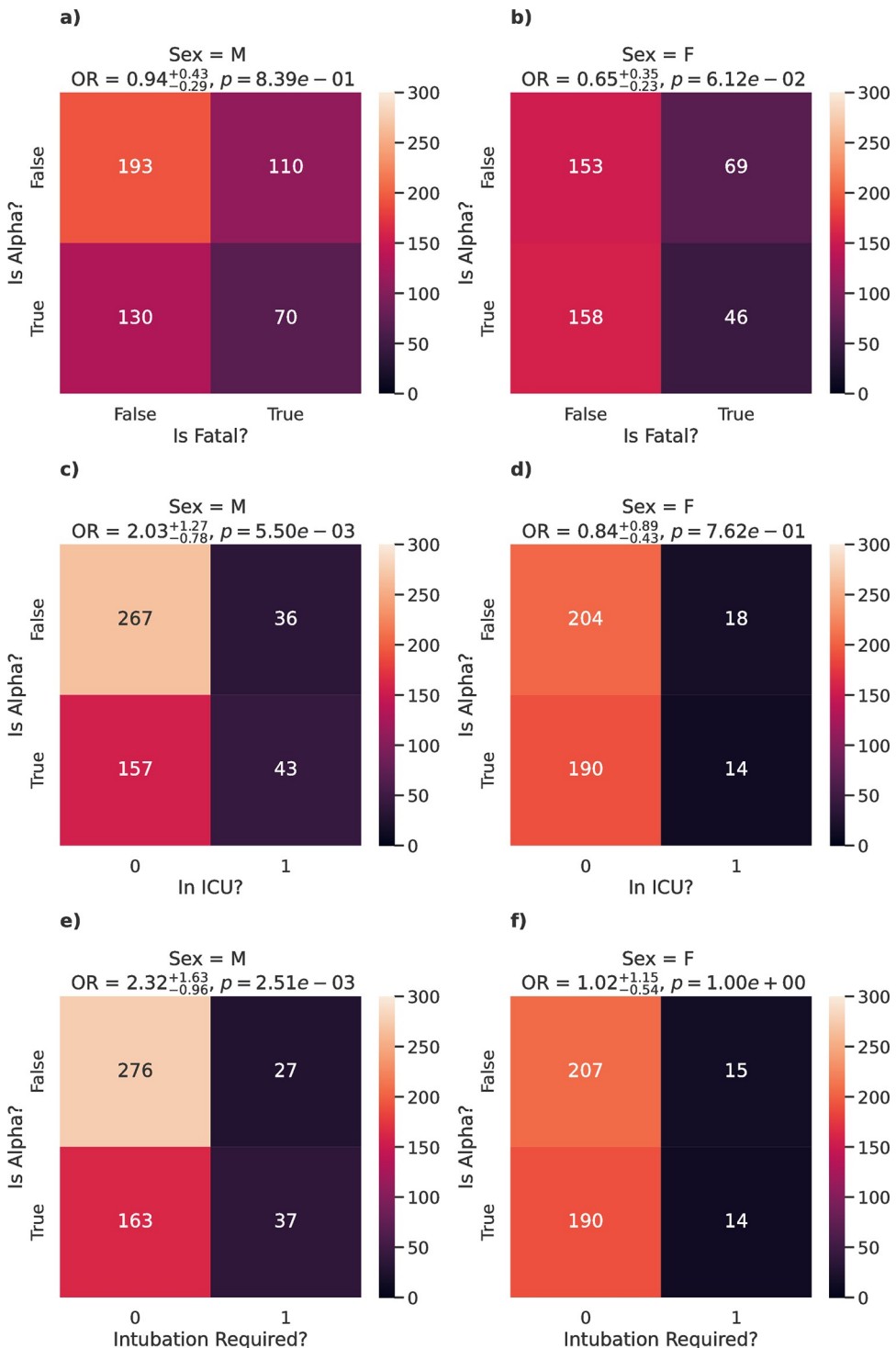

**Fig 4. Effect of the Alpha variant on disease severity for male and female patients.** Heatmaps exploring the changes in relative risk of death (a-b), ICU admission (c-d) and intubation (e-f) for male and female patients when comparing Alpha cases to non-Alpha cases. The odds ratio and p-values $p$ between the classes are shown above each panel.

**Table 3. Association between patient variables and SARS-CoV-2 mutations.**

| Feature | Association Strength Metric | A | p |
|---|---|---|---|
| Length of Stay | Correlation Ratio | 0.25 | 5.89E-16 |
| Admission NEWS2 Score | Correlation Ratio | 0.25 | 1.01E-15 |
| Number of Pre-Existing Conditions | Correlation Ratio | 0.24 | 3.15E-06 |
| Maximum NEWS2 Score | Correlation Ratio | 0.23 | 1.27E-09 |
| Time to Discharge/Death from Diagnosis | Correlation Ratio | 0.23 | 5.93E-06 |
| Admission Specialty | Correlation Ratio | 0.16 | 0.00E+00 |
| Ethnic Origin | Cramer's V | 0.15 | 0.00E+00 |
| Malignancy Indicator | Cramer's V | 0.14 | 1.57E-13 |
| Swab Location | Cramer's V | 0.14 | 0.00E+00 |
| Ward | Cramer's V | 0.14 | 0.00E+00 |
| Renal Disease Indicator | Cramer's V | 0.13 | 4.08E-11 |
| Sex | Cramer's V | 0.12 | 7.48E-08 |
| COPD Indicator | Cramer's V | 0.11 | 7.01E-08 |
| Location Category | Cramer's V | 0.11 | 1.02E-35 |
| Admission Age | Cramer's V | 0.10 | 1.94E-283 |

Table of variables with statistically significant links with the single nucleotide polymorphisms and other mutations of the SARS-CoV-2 genome, showing the association strength $A$ (based on Cramer's V for categorical-categorical relationships and Correlation Ratio for categorical-continuous relationships) and p-value $p$ of the relationship.

## SARS-CoV-2 mutations associated with severe COVID-19

As the SARS-CoV-2 virus has mutated over time, a number of key mutations have been identified, particularly in the spike protein of the virus. We used our dataset to identify whether any specific mutations or clusters of mutations could be identified that might be associated with an increased risk of a negative outcome, thus acting as predictors of outcome in future cases. Features from the joint outcomes and mutation dataset that showed statistically significant relationships with single nucleotide polymorphisms (SNPs) and deletions of the SARS-CoV-2 genome are shown in Table 3. We found no statistically significant link between the SNPs and any of our chosen indicators of disease severity (death, ICU admission, or intubation). Interestingly, we found a moderately weak link between the mutations and the NEWS2 score, both at admission ($A$ = 0.25, p = 1.01e-15) and the maximum recorded score ($A$ = 0.23, p = 1.27e-09). This may indicate that some mutations impact acute physiological status, but not enough to directly result in a severe case of COVID-19. This hypothesis is supported by the weak association between the mutations and the patient's length of stay ($A$ = 0.25, p = 5.89e-16), indicating a weak link between the types of mutations found in patients who experienced symptoms of COVID-19 for longer periods of time. Also interestingly, we see that the mutations were associated with the number of pre-existing conditions the patient has ($A$ = 0.24, p = 3.15e-06) as well as whether the patient has cancer ($A$ = 0.14, p = 1.57e-13), renal disease ($A$ = 0.13, p = 4.08e-11), or COPD ($A$ = 0.11, p = 7.01e-08), although these associations are quite weak. We also see weak associations between the mutations and the demographics of the patient, in particular their ethnic origin ($A$ = 0.15, p < 1.00e-300), sex ($A$ = 0.12, p = 7.48e-08) and age ($A$ = 0.10, p = 1.94e-283). Similar weak associations are also seen with the locations of the patient, particularly the admission speciality ($A$ = 0.16, p < 1.00e-300), the location where the patient was swabbed ($A$ = 0.14, p < 1.00e-300) and the ward the patient was located after admission to the hospital ($A$ = 0.14, p < 1.00e-300), likely arising as a result of nosocomial spread of the virus within wards.

## Machine learning (ML) and artificial neural network (ANN) analysis of mutations and comorbidity risk-factors associated with disease severity

To further explore the role of viral mutations of SARS-CoV-2 in severity of disease in COVID-19, we utilised machine learning approaches to identify mutations with a possible role in determining patient outcomes. Nine machine learning algorithms and one deep-learning neural network method were tested and ranked according to their accuracy (Fig 5A), with a binary outcome of death (outcome = 1) or no death (outcome = 0) following escalation of care to the ICU. Of these, the XGradient Boosted (XGBoost) and the Multi-Layer Perceptron Artificial Neural Network (MLP-ANN) approaches produced the best results, with the MLP-ANN model resulting in slightly improved accuracy (76.2%; Fig 5B, right) compared to XGBoost (74.6%; Fig 5B, left).

Comparison of macro-average metric scores between the XGBoost and MLP-ANN models are shown in Fig 5B, with the breakdown for the different outcomes shown in Fig 5C. In both models, Precision and Recall were high in discrimination of COVID-19 patients with greater survival probability (low-risk patients) following their admission to intensive care units, but low for discrimination of high-risk patients. The MLP-ANN model showed a higher Recall (100% vs 92%), but lower Precision (74% vs 78%) for identification of low-risk patients compared to the XGBoost model. Whilst potentially unsuitable for identification of high-risk patients, this model may potentially offer an approach for exclusion of low-risk patients, allowing for the remaining cohort to be observed as potentially high risk, with resources prioritised for more intensive clinical surveillance, management, and attention.

The ranking of SNPs, deletions, and clinical criteria in the order of importance to the model (from top to bottom), based on the Shapley additive explanation (SHAP) values, is shown for the XGBoost model (Fig 5D) and the MLP-ANN model (Fig 5E). For mutations identified as the top predictors variables, the numeric ID indicates the nucleotide position of the nucleotide change relative to the reference SARS-CoV-2 genome, and the gene domain in which the mutation is located is highlighted (S4 Fig).

The feature importance of the predictor variables is different for the XGBoost model compared to the MLP-ANN model, with individual genomic features in the XGBoost model typically showing higher mean SHAP values than those in the MLP-ANN model. Another striking difference is in the ranking of clinical variables, which represent the top ranked features in the XGBoost model, but appear less significant than a number of mutations (in particular deletions) in the MLP-ANN model. Renal disease, heart disease and diabetes feature for both models, with risk factors such as history of hypertension, COPD, prior history of malignancy, and asthma prominent in the XGBoost model, but not in the MLP-ANN model.

One notable similarity between the two models, SNP 23403, corresponds to an A->G mutation at site 23,403, resulting in an Aspartic Acid to Glycine amino acid substitution (D614G) within the spike (S) protein domain. This SNP, seen in the top 20 for both models, became rapidly dominant globally due to increased viral fitness and higher viral loads [42–45]. It became fixed in the population after the first wave in the UK, with 0% of cases showing the Glycine residue in January 2020, rapidly increasing to 70% of cases in May 2020 [45]. It is therefore likely that association of this SNP with severity is closely linked with temporal developments of the pandemic, with significant improvements in treatment options and vaccine developments from the second wave onwards.

Whilst the spike protein has been linked with increased viral load and fitness [46], and thus may represent an obvious source for identification of mutations linked with disease severity due to its role in host cell receptor binding, the majority of the identified mutations actually seem to lie in other regions of the viral genome. In particular, for the XGBoost model, the

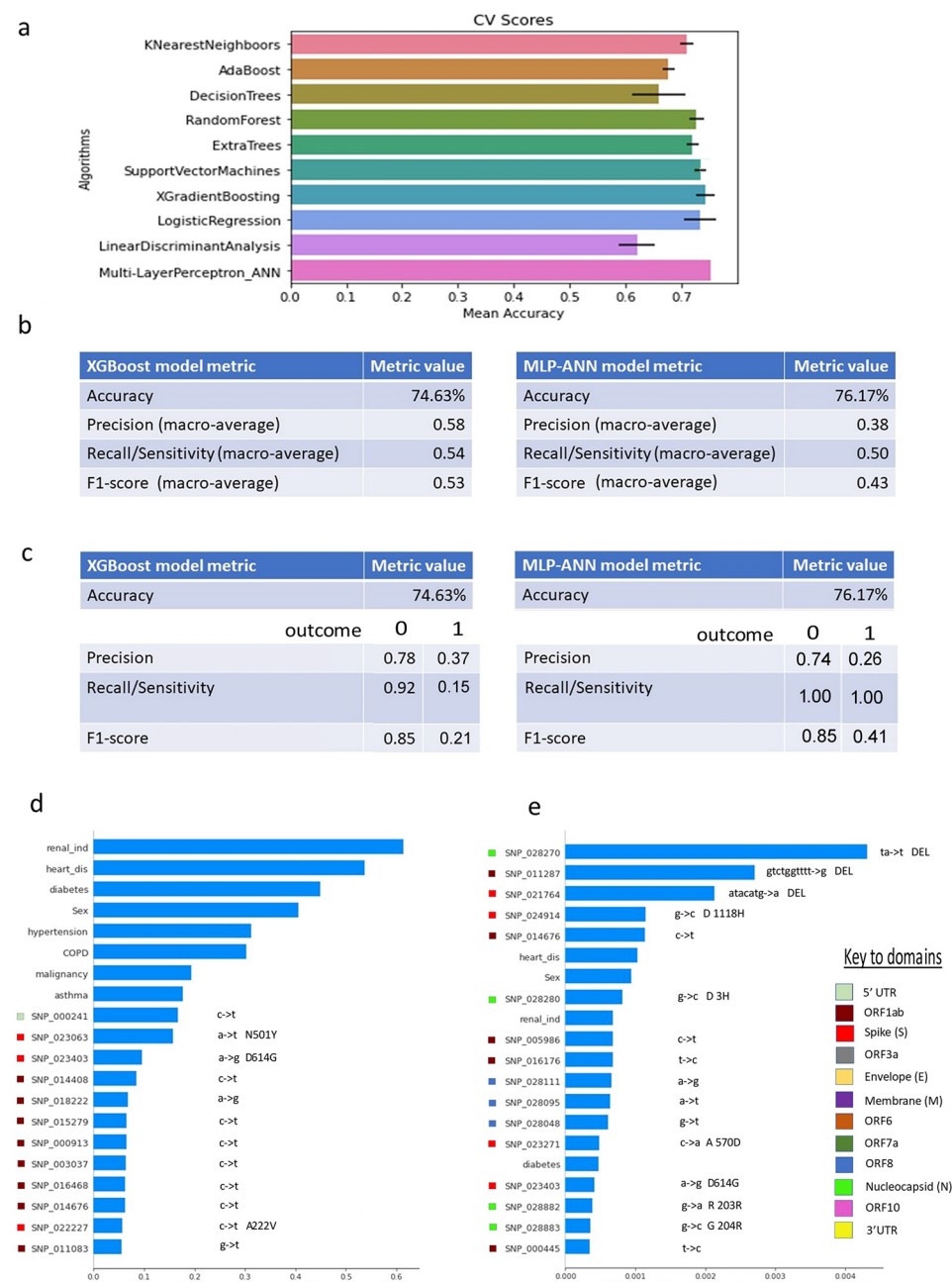

**Fig 5. Machine learning and artificial neural network analysis of mutations in the SARS-CoV2 genome isolated from patients with COVID-19.** (a) Screening and comparison of nine machine-learning and one deep-learning multi-layer perceptron artificial neural network (MLP-ANN) method showing mean percentage accuracy of prediction of outcome; death (1) or no death (0). Error bars shown represent standard deviation of accuracy run over 200 estimators (machine-learning) or 300 epochs for neural-net analysis. (b) Macro-average (where metrics for outcomes 1 and 0 are averaged) metrics comparison between the XGBoost and the MLP-ANN method. The MLP-ANN gave better macro-average accuracy metrics. (c) Individual metrics shown for outcomes 1 (death) and 0 (no death), compared between XGBoost and MLP-ANN models. The MLP-ANN gave better accuracy metrics and overall Sensitivity metrics. (d) Ranking of SNPs and clinical parameters in order of importance, displayed according to their SHapley Additive exPlanations (SHAP) values in their predictive ability in the respective models. The location of the SNPs in relation to the SARS-CoV2 genome is based on mapping to the reference genome (Wuhan-Hu-1, GenBank, MN908947.3) and shown using colour-coded keys. Also shown are the nucleotide changes noted in relation to each SNP.

majority of other significantly associated mutations were identified within the ORF1ab gene. These include synonymous SNPs such as SNP 913 (C->T), SNP 3037 (C->T), SNP 14408 (C->T), SNP 16468 (C->T), as well as non-synonymous SNPs such as SNP 23603, an A->T transition resulting in a change from Asparagine to Tyrosine at amino acid (AA) 501 in the spike domain identified in Alpha cases.

Interestingly, whilst no deletions are present in the top ranked features for the XGBoost model, they represent the top three mutations in the MLP-ANN model. All three deletions (SNP 28270, a 1 bp frameshift deletion in the Nucleocapsid (N) domain; SNP 21764, a 6 bp deletion in the spike (S) domain; SNP 11287, a 9 bp deletion in ORF1ab) are specific to the Alpha lineage B.1.1.7, and clearly delineate Alpha from non-Alpha cases. Similarly, the majority of the remaining mutations identified by the MLP-ANN model appear to be highly specific to the Alpha lineage, indicating that this model primarily identifies the presence of the Alpha lineage as being associated with disease severity. As with SNP 23403, this is likely linked to temporal development in treatment options for those with Alpha later in the pandemic compared to cases with severe symptoms in earlier waves.

## Discussion

As the world returns to a more normal state after being plunged into a global pandemic, many questions remain to be answered about COVID-19. In particular, it is still not well understood exactly which factors are most associated with the likelihood of an individual suffering from the most significant negative outcomes, including long-term post-COVID-19 respiratory issues ("long COVID"), requirement for invasive mechanical intubation, admission to ICU, or even death.

In this study, clinical data were linked to viral genomic data from patients seen across an acute NHS Trust on the south coast of the UK. This data resource was used to explore potential links between severe outcomes and viral subtype, patient demographics, and clinical history, to further understand factors that may influence patient responses to the virus. Overall, this study found no strong factors associated with severe cases of COVID-19, instead showing weak influence from myriad factors including age, sex, and existence of pre-existing conditions.

Of course, certain pre-existing conditions are more likely than others to directly influence COVID-19 illness. For example, given that cataracts are typically seen in older individuals, many of those most clinically vulnerable for severe COVID-19 outcomes may suffer from cataracts, with one-fifth of patients awaiting cataract surgery found to be at high risk of severe disease or death from COVID-19 in a 2022 study [47]. However, whilst a serious malady and a leading cause of preventable blindness, suffering from cataracts is itself unlikely to have a significant bearing on the severity of COVID-19 pneumonia. The context of the comorbidity in relation to the subsequent pathology of SARS-CoV-2 pathophysiology is important, since the primary target organs are the lungs, and pathophysiological progression may require mechanical ventilation in areas where high-dependency or intensive care is offered.

One of the frequently observed disease progressions in COVID-19 is the persistence of micro-coagulopathy, where tiny clots systemically occlude capillaries, such as in the glomeruli [48]. Thus, a patient with a pre-existing compromised renal function, or those with pre-existing cardiac dysfunction (especially previous coronary ischaemia) might show poor recovery trajectories in hospital. It is thus clear that certain pre-existing conditions, particularly renal and heart disease, may make an individual more likely to suffer from severe complications with COVID-19 and have been previously identified as risk factors [49]. Indeed, as shown in Table 2, renal disease ($A = 0.15$), heart disease ($A = 0.14$), and cancer ($A = 0.11$) were identified

as being significantly associated with the likelihood of death. Such patients should therefore continue to be monitored closely, to observe signs of deterioration.

It is worth noting however, that the absolute increase to risk is low based on our data, and a relatively large proportion of those analysed suffered from heart disease (52.7%), renal disease (37.2%), and cancer (11.6%) (Table 1). Indeed, only 13.8% of patients in our dataset had no pre-existing condition at all, highlighting a significant selection bias in the data. There is also a selection bias with admission age, with 79.3% of patients aged 60 and above and a median age of 76. These biases are likely closely related, since older patients typically experience a higher proportion of comorbidities compared to younger age groups [9]. These selection biases may impact other association scores, potentially resulting in underestimated scores for pairwise associations with admission age and comorbidities.

These data also highlight that acute physiological derangement of the patient is linked to severe COVID-19, indicated by a moderate-strong association between the maximum NEWS2 score and whether the patient died within 30 days of diagnosis ($A = 0.43$), was admitted to ICU ($A = 0.21$), or required invasive mechanical intubation ($A = 0.21$) (Table 2). The NEWS2 score reports on a constellation of dynamically changing (particularly within an acute setting) clinical features, but is a simple to calculate metric to identify and address patient deterioration [32,33], and has been previously identified as a potential screening tool for severe patient outcomes [50–52]. However, a UK multicentre study identified poor to moderate discrimination of medium-term COVID-19 outcomes from NEWS2 scores and age alone, calling into question its use as a screening tool [53]. A common observation with COVID-19 is of mild phenotypes deteriorating towards severe phenotypes (resulting in an increased NEWS2 score) as a result of the respiratory distress caused by COVID-19 pneumonitis. In comparison, the NEWS2 score given to a patient on admission shows no significant association with death, suggesting that it is unlikely to represent a significant predictive factor for COVID-19 severity. A weak association is seen between admission NEWS2 and both ICU admission ($A = 0.24$) and intubation ($A = 0.24$), but given that the NEWS2 score is often a tool used to determine whether a patient has deteriorated sufficiently to require intubation or ICU treatment, this is perhaps unsurprising. Length of stay also showed moderate associations with ICU admission ($A = 0.30$). There are several risk factors which become apparent with an increased length of hospital stay, for instance the likelihood of the patient being on prolonged prescription of several non-routine medications. These include medication for prevention of venous thromboembolism (heparin and other anticoagulants), medications to aid somnolence at night (sleep medication is frequently requested by the elderly while at hospital due to unfamiliar disturbing noises at night in a busy clinical environment), antibiotics, anti-anxiety medication, medication to help bowel movements (due to prolonged bed-rest and immobility), medication to off-load water retention from immobility (again from prolonged bed-rest) and pain medication.

Other factors showing moderate association with ICU admission included the location category of their treatment ward ($A = 0.66$), the specific ward number ($A = 0.49$), and the ward in which their COVID-19 test swab was collected ($A = 0.43$). Patients at PHU are triaged and risk-stratified on admission, and the location of the clinical setting that they are initially taken to for treatment would reflect the clinical need for specialist services, equipment or staff-training levels distributed within a particular sector within the hospital. Such a sector is typically populated with a high number of patients needing high-dependency care and treatment. Since aerosolization of the virus is a potential and proven risk, along with the potential for direct transmission from person to person, nosocomial spread within such high-dependency care units results in increased cases within these areas. It is therefore likely that associations of outcomes with location-related data are a result of localized outbreaks, resulting in cases with shared mutation patterns between patients who share similar treatment and comorbidity

characteristics. Indeed, nationally over 15% of all cases have been estimated as having been hospital acquired in the first wave in the UK [54], with up to 20% of infections in inpatients and 73% in HCW due to nosocomial transmission [55]. It has been suggested that up to 80% of nosocomial infections were caused by only 20% of patients due to "super-spreader" events [14], with such rapid outbreak dynamics having been previously characterised in at least one outbreak at PHU [56].

One key question to address as new variants of SARS-CoV-2 continue to arise is the effect on severity of the disease as a result of new variants. Whilst the data described here do not span the emergence of variants such as Delta and Omicron, they do represent the emergence and subsequent rapid expansion of the first VOC, Alpha (B.1.1.7). Increased prevalence of Alpha in the local region led to increased transmission of a range of currently circulating variants within the hospital [56]. Interestingly, Table 1 shows that the rate of severe cases amongst Alpha cases (36.9%) was actually slightly lower than amongst non-Alpha cases (40.2%), suggesting that Alpha cases may present a lower risk of severe outcomes in our dataset compared to other variants (Table 1). However, whilst lineage was weakly associated with ICU admission ($A = 0.18$), we otherwise saw no statistically significant links between lineage and death, intubation, nor case severity in general (Table 2). In addition, we identified changes to the odds of severe outcomes for cases of the Alpha VOC compared to other circulating variants for certain sub-populations. In particular, whilst the risk of severe outcomes was significantly higher amongst males compared to females in general (OR = 1.81), which is consistent to previous studies [57–61], the overall risk showed a moderate (although not significant) reduction in cases of the Alpha variant when compared to other cases for females (OR = 0.65) but not males (OR = 1.17). Looking specifically at our three severity indicators identified a mild non-significant decreased risk for mortality amongst females, but in contrast showed a significant increase in risk in males for admission to ICU (OR = 2.03) and intubation (OR = 2.32).

Overall, these results suggest that whilst the Alpha variant had no significant impact on COVID-19 severity overall, specific subgroups of the population may be more or less impacted by specific variants of the COVID-19 virus over others. Differences in the impact of SARS-CoV-2 infections between males and females has been suggested to result from differences in the expression of angiotensin converting enzyme (ACE2) receptors [62]. Indeed, circulating ACE2 levels have been shown to be higher in men, as well as in those with diabetes and pre-existing cardiovascular conditions [63]. The study of Stirrup *et al* [26], a large-scale multi-centre study in the UK, also found that overall hazard of mortality and ICU admission were not significantly affected in cases of Alpha compared to other lineages, but that sex-specific effects may be present. Interestingly, however, they showed that it was women specifically that showed increased risk of mortality and ICU admission in their cohort. Increased mortality appeared to be specific to those 70 years and above, with a slight decrease seen in 50–69 year olds. One possible explanation for this discrepancy may therefore be in differences in the age profiles of those included in the two studies. Another possible explanation may be that our dataset contains cases from across the entire course of the pandemic, including the first UK wave where risk of severe outcomes was higher as a result of a lack of identified treatment and vaccine options. Indeed, a recent large-scale study of 30 million people in the UK showed that risk of severe COVID-19 outcomes is reduced as a result of ongoing vaccine programs [31]. However, our result remains when focussing only on cases from September 2021, indicating that wave 1 patients do not affect the outcome data. It is worth also noting that whilst the Alpha variant data are largely homogenous, significant heterogeneity exists in the non-Alpha data, with cases coming from 46 distinct lineages in these data. Another difference may be with respect to the population under consideration, since Stirrup *et al* was a multi-centre study, primarily from hospitals within London (although did include data from the nearby city

of Southampton). Both studies however point towards the role of Alpha in disease severity being context specific and mild overall.

Linkage of WGS and clinical data represents a powerful approach for assessment of the effects of Alpha on severity, in comparison to studies which used surrogate measures such as S-gene target failure (SGTF) in qPCR tests to differentiate Alpha from other lineages. Indeed, other studies based on community testing and SGTF have shown conflicting results, with studies showing increased risks of Alpha, but no difference in the effects of Alpha on mortality [64,65] or ICU admission [65] between male and female cases. Thus, the evidence for increased severity of the Alpha variant of concern remains inconclusive [66]. Beyond the role of VOCs in determining disease severity, we sought to identify potential mutations or mutation clusters associated with patients who suffered severe outcomes. Whilst we found no significant link between lineage and overall severity, we did find a weak link between the mutation type and the NEWS2 scores given to the patient at admission ($A$ = 0.25) and the maximum score assigned ($A$ = 0.23) (Table 3). Whilst this may indicate that there are mutations associated with patient health and physical derangement, it is also possible that such links relate to nosocomial transmission of the disease amongst clinically vulnerable patients, as previously discussed. This is further suggested given that the association is mostly enriched for mutations associated with non-severe outcomes.

To explore this in more detail, we utilised a range of machine learning models with individual mutations encoded alongside other patient factors, to further explore associations with patient mortality. Deep learning models have previously been developed for use in the diagnosis and screening of COVID-19 through interrogation of CT and chest X-ray images [67]. The two models with highest accuracy, XGBoost and MLP-ANN, were compared to identify features most linked with mortality. Renal disease, heart disease and diabetes feature for both models, with risk factors such as history of hypertension, COPD, prior history of malignancy, and asthma prominent in the XGBoost model, but not in the MLP-ANN model. The stochastic nature of algorithms such as XGBoost and MLP-ANN models means a degree of randomness exists, contrasted with deterministic algorithms such as linear regression or logistic regression-based models. Regardless, it is clear that comorbidities are amongst the features most closely associated with disease severity. Whilst the XGBoost model identified comorbidity status and sex as being most predictive of severity (Fig 5D), the MLP-ANN identified a number of deletions as being the features with the most impact (Fig 5E). These deletions were all specific to the Alpha variant B.1.1.7, including the Δ69–70 deletion on the Spike protein responsible for SGTF in qPCR testing for Alpha [68–71].

These deletions are therefore likely identified by the model as surrogates for Alpha vs non-Alpha cases. Whilst this may indicate that Alpha may be associated with mortality, this is not borne out when looking at male and female cases individually (Fig 4). This is therefore likely the result of non-Alpha lineages primarily representing cases from earlier in the pandemic, but may also be linked to selection bias due to Alpha being over-represented in these data. Similarly, the well documented D614G mutation was identified by both models, which was introduced at low levels during the first wave of infections in the UK, but became dominant and fixed in the population in subsequent waves [45]. This mutation is also linked with the temporal nature of the pandemic, with severity often being worse in earlier waves due to the lack of treatment options, reduced testing and interventions, and lack of vaccine program. It is therefore likely that these mutations are highlighting differences between cases early and later in the pandemic, rather than inherently having a functional role in increasing disease severity.

Overall, our analysis indicates that there are no clear strong factors that determine severe outcomes from COVID-19 (mortality, ICU admission or intubation). Whilst we detected a number of significant associations, most were mild and could be explained due to conflation

with either general patient health, their location within the hospital, or changes in our treatment capabilities for the disease throughout the pandemic. It has been previously shown that comorbidities such as cancer, renal disease and heart disease are linked to negative outcomes, particularly mortality [49]. Also, whilst it is interesting to note that the NEWS2 score showed significant association with disease outcomes, these are not suitable for prediction of outcomes as discussed above. Similarly, the characteristics of the viral variant at the root of the infection is unlikely to present a suitable predictive tool for determining disease outcomes. Whilst there was some evidence of effects on severity from the Alpha variant compared to other circulating variants, the effect was inconsistent, with both increase and decrease in severity seen, sometimes at odds with previous studies.

Whilst this study focuses on only the Alpha variant, and thus cannot draw conclusions for further VOCs such as Delta and Omicron, these results suggest that within these data the introduction of the Alpha variant did not have a significant impact on severity of the disease. Of course, these data represent only a limited population, with 929 patient samples from a single hospital site. One other key limitation of this study is that the demographics of the patient cohort are skewed for those of the local area, in particular with over 75% of those in the study being of a white background (Table 1). These results may therefore not be generalisable to the population as a whole. However, despite these limitations, our study represents a useful and in-depth interim exploration of the effects on disease severity in response to both clinical measures and viral genomics. Recently, a large-scale analysis of over 1 million patients in England showed lower or similar risks of death, hospital admission and hospital attendance between the BA.1 and BA.2 Omicron variants [30], matching our observation that emerging SARS-CoV-2 variants do not result in more severe outcomes for patients.

Since our data indicate that virus genomics have limited impact on disease severity, it is likely that understanding of those most susceptible to severe outcomes when infected by SARS-CoV-2 (beyond clinically vulnerable individuals) will come from studies such as the GenOMICC study in the UK (https://genomicc.org/about/), which aim to understand the interaction between virus and host, and explore genetic factors in humans that dictate disease outcomes. Indeed, multiple studies have already been conducted identifying potential susceptibility loci in the human genome that may put patients at increased risk of death or other severe outcomes, including mutations in genes linked to immune response, blood clotting and mucus production [72–75]. In particular, a recent study using machine learning approaches such as XGBoost identified variants from whole exome sequencing associated with severe COVID-19 [76]. These data identified associations between age, gender, and 16 variants linked to immune system and inflammatory processes able to predict severe outcomes with high accuracy. Such studies will help to further understand the factors that predispose individuals to severe outcomes from SARS-CoV-2 infection.

As society accustoms itself to a "new normal" way of life, we are learning to live with endemic COVID-19. New variants will continue to emerge, and it is therefore imperative that we learn what we can from existing data. It is particularly important for us to understand how the most severe disease cases arise, in the hope that we may target such cases specifically and early. Studies like this which combine clinical and laboratory data, will thus be essential to that task.

## Conclusion

Whilst many risk factors for severe COVID-19 have been identified, the precise mechanisms resulting in severe outcomes for those infected by SARS-CoV-2 (including admission to ICU, the need for mechanical ventilation, and mortality) remain poorly understood. In this study,

we aimed to combine genomic sequencing data of SARS-CoV-2 viral variants with an extensive database of patient records to further understand those factors most associated with severe outcomes. In particular, we were interested to understand the precise role played by mutations in the virus itself, and whether infection with certain variants or viruses with specific mutations might be more likely to cause severe disease. Whilst patient outcome was weakly associated with factors linked with acute physiological status and human genetics, including age, sex and pre-existing conditions, our data suggest that severity risk is not significantly impacted by specific mutations in SARS-CoV-2. It is therefore likely that risk of severe outcomes results from a combination of patient health and innate genetic predisposition. Thus, whilst studies such as ours significantly further our understanding of the pathophysiology of the virus, ongoing studies exploring the role of host genetics on disease progression will continue to disentangle the complex factors that might increase risk to those infected with SARS-CoV-2.

## Supporting information

**S1 Fig. Sample flowchart.** Flowchart of filtering steps for the final joint dataset.
(TIF)

**S2 Fig. Machine learning performance metrics.** Performance metrics during the training and validation and the architecture of the MLP-ANN model. (a) The sequential method in Tensorflow v2.8 was used, incorporating the Adam optimization algorithm for stochastic gradient descent for training of deep learning models. Parameters used were a learning rate of 0.0001, with beta_1 = 0.9 and beta_2 = 0.799. Following initial stages of 10,000 epochs, the model was refined and optimised for the appropriate number of nodes and hidden layers, and an "early stopping" protocol was incorporated to stop training once the model performance stopped improving. This was determined using a concurrent evaluation of cross-validation loss remaining similar over 20 epochs, and ensured minimal over-fitting and improving computing time. The two graphs here show close convergence and agreement between the train and validation sets of the MLP-ANN model. (b) The final architecture of the MLP-ANN model. The model contained 3 hidden layers (with 700, 700 and 10 nodes each), and a final output layer containing two nodes to pipe the categorical outcomes of 0 (no-death) and 1 (death). The number of optimal nodes were optimised over several runs of model building and hyperparameter optimisation steps. The final layer used "softmax" as the activation step, which scales numbers/logits into probabilities. The activation steps for the hidden layers were ReLU, used specifically to address the problem of vanishing gradients in deep-learning models. Dropout regularization was employed to reduce overfitting of the model, where different sets of neurons are dropped from the architecture, giving an overall result akin to training and optimizing multiple neural networks simultaneously.
(TIF)

**S3 Fig. Pairwise association score heatmap.** Heatmap showing pairwise association scores (based on Cramer's V for categorical-categorical relationships, Spearman's Rank for continuous-continuous relationships and Correlation Ratio for categorical-continuous relationships) between all variables in the patient outcomes dataset. Only statistically significant results are shown, with any association with $p > 0.05$ shown as grey. See S1 Table for description of data points.
(TIF)

**S4 Fig. SARS-CoV-2 genome schematic.** Schematic organisation of the SARS-CoV2 genome and the coordinates of nucleotide positions marking the boundaries of the various viral domains. The schematic diagram was generated using VectorNTI (V11). The wild-type

SARS-CoV2 genome sequence was obtained from Genbank (Wuhan-Hu-1, GenBank, MN908947.3). The arrows indicate the direction of translation of the gene (5' to 3').
(TIF)

**S1 Table. Primary data point descriptions.** Description of primary data points in the combined data set.
(XLSX)

**S1 File. COG-UK consortium.** Full list and affiliations for COVID-19 Genomics UK (COG-UK) Consortium members.
(DOCX)

## Acknowledgments

The authors would like to thank all of the NHS patients and staff who supported this work to be undertaken throughout the pandemic. We also thank the COG-UK Consortium and its members for their constant support and assistance, and the UK National Institute for Health Research Clinical Research Network (NIHR CRN) for their support of the work. In addition, we would also like to thank everybody at the University of Portsmouth and PHU who worked behind the scenes to ensure that this work could proceed, including those in management, finance, purchasing, and procurement teams at the Faculty of Science and Health.

## Author Contributions

**Conceptualization:** Andrew Lundgren, Or Graur, Anoop J. Chauhan, Samuel C. Robson.

**Data curation:** Max Foxley-Marrable, Leon D'Cruz, Paul Meredith, Sharon Glaysher, Sally Lumley, James McNicholas, David Prytherch, Samuel C. Robson.

**Formal analysis:** Max Foxley-Marrable, Leon D'Cruz, Samuel C. Robson.

**Funding acquisition:** Andrew Lundgren, Or Graur, Anoop J. Chauhan, Samuel C. Robson.

**Investigation:** Sharon Glaysher, Angela H. Beckett, Salman Goudarzi, Christopher Fearn, Kate F. Cook, Katie F. Loveson, Hannah Dent, Hannah Paul, Scott Elliott, Sarah Wyllie, Allyson Lloyd, Kelly Bicknell, James McNicholas, David Prytherch, Samuel C. Robson.

**Methodology:** Max Foxley-Marrable, Leon D'Cruz, Sharon Glaysher, Angela H. Beckett, Samuel C. Robson.

**Project administration:** Anoop J. Chauhan, Samuel C. Robson.

**Resources:** Sharon Glaysher, Scott Elliott, Sarah Wyllie, Allyson Lloyd, Kelly Bicknell.

**Software:** Max Foxley-Marrable, Leon D'Cruz, Samuel C. Robson.

**Supervision:** Andrew Lundgren, Or Graur, Samuel C. Robson.

**Visualization:** Max Foxley-Marrable, Leon D'Cruz, Samuel C. Robson.

**Writing – original draft:** Max Foxley-Marrable, Leon D'Cruz, Samuel C. Robson.

**Writing – review & editing:** Max Foxley-Marrable, Leon D'Cruz, Paul Meredith, Sharon Glaysher, Angela H. Beckett, Salman Goudarzi, Christopher Fearn, Kate F. Cook, Katie F. Loveson, Hannah Dent, Hannah Paul, Scott Elliott, Sarah Wyllie, Allyson Lloyd, Kelly Bicknell, Sally Lumley, James McNicholas, David Prytherch, Andrew Lundgren, Or Graur, Anoop J. Chauhan, Samuel C. Robson.

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
