## [Decision Letter · Decision Letter 0]

2 Jan 2023

PONE-D-22-31171Combining viral genomics and clinical data to assess risk factors for severe COVID-19 (mortality, ICU admission, or intubation) amongst hospital patients in a large acute UK NHS hospital TrustPLOS ONE

Dear Dr. Robson,

Thank you for submitting your manuscript to PLOS ONE. After careful consideration, we feel that it has merit but does not fully meet PLOS ONE’s publication criteria as it currently stands. Therefore, we invite you to submit a revised version of the manuscript that addresses the points raised during the review process.

We look forward to receiving your revised manuscript.

Kind regards,

Divyansh Agarwal

Academic Editor

PLOS ONE

Journal Requirements:

"The authors would like to thank all of the NHS patients and staff who supported this work to be undertaken throughout the pandemic. We also thank the COG-UK Consortium and its members for their constant support and assistance, in particular in funding this project under their Internal Principal Investigator Research Funding Scheme. We thank the UK National Institute for Health Research Clinical Research Network (NIHR CRN) for their support of the work."

"This work was primarily funded by the COVID-19 Genomics UK (COG-UK) consortium (https://www.cogconsortium.uk/), under their Internal Principal Investigator Research Funding Scheme. COG-UK is supported by funding from the Medical Research Council (MRC; https://www.ukri.org/councils/mrc/) part of UK Research & Innovation (UKRI; https://www.ukri.org/), the National Institute of Health Research (NIHR; https://www.nihr.ac.uk/) [grant code: MC_PC_19027], and Genome Research Limited, operating as the Wellcome Sanger Institute (https://www.sanger.ac.uk/). The authors acknowledge the use of data generated through the COVID-19 Genomics Programme funded by the Department of Health and Social Care (DHSC; https://www.gov.uk/government/organisations/department-of-health-and-social-care). The views expressed are those of the author and not necessarily those of the Department of Health and Social Care or PHE or UKHSA. MFM, AL, and OG were also supported by a UKRI Science & Technology Facilities Council (STFC) Impact Accelerator Account awarded to the Institute of Cosmology and Gravitation (ICG) at the University of Portsmouth. Additional funding for the project came from the University of Portsmouth Faculty of Science and Health (https://www.port.ac.uk/about-us/structure-and-governance/organisational-structure/our-academic-structure/faculty-of-science-and-health), and the Wessex Academic Health Sciences Centre (AHSC; https://wessexahsn.org.uk/). In addition, SCR and AHB are funded by Research England’s Expanding Excellence in England (E3) Fund. The primary funders had no role in study design, data collection and analysis, decision to publish, or manuscript preparation."

"Scott Elliot and Salman Goudarzi currently work for QIAGEN, UK. The remaining authors declare that no competing interests exist."

We note that one or more of the authors are employed by a commercial company: QIAGEN, UK.

4. We noted in your submission details that a portion of your manuscript may have been presented or published elsewhere. "The whole genome sequencing data of SARS-CoV-2 samples used in this manuscript have been submitted to the COVID-19 Genomics UK Consortium database, and used to underpin a wide range of publications across the Consortium that have used the entire database to explore cases from across the UK. In addition, we have submitted a manuscript using these sequencing data that is currently undergoing review with Frontiers in Cellular and Infection Microbiology: Molecular Viral Pathogenesis. That manuscript focusses on using these data to identify shared infections and nosocomial transmission chains within the hospital amongst staff and patients, whilst the submitted manuscript combines viral genomic data with a range of other clinical data to assess risk factors for disease severity. These are thus very different studies, using the same dataset to address distinct but important questions, and thus do not constitute dual publication." Please clarify whether this publication was peer-reviewed and formally published. If this work was previously peer-reviewed and published, in the cover letter please provide the reason that this work does not constitute dual publication and should be included in the current manuscript.

6. One of the noted authors is a group or consortium The COVID-19 Genomics UK (COG-UK) consortium. In addition to naming the author group, please list the individual authors and affiliations within this group in the acknowledgments section of your manuscript. Please also indicate clearly a lead author for this group along with a contact email address.

7. Your ethics statement should only appear in the Methods section of your manuscript. If your ethics statement is written in any section besides the Methods, please move it to the Methods section and delete it from any other section. Please ensure that your ethics statement is included in your manuscript, as the ethics statement entered into the online submission form will not be published alongside your manuscript. 

Additional Editor Comments:

Dear Authors,

Thank you for submitting your work on combining viral genomics and clinical data to assess risk factors for severe COVID-19. As you will see, the reviewers felt that the work is meritorious but will benefit from some revisions. Our editorial assessment is along similar lines. We are excited by the importance of the question being asked, and would welcome a revised version of the manuscript. I encourage you to please submit a revised version of the manuscript with a point-by-point response to the reviewer's comments.

Please don't hesitate to reach out with any questions.

Best wishes for a Happy New Year.

Reviewers' comments:

Reviewer's Responses to Questions

**Comments to the Author**

1. Is the manuscript technically sound, and do the data support the conclusions?

Reviewer #1: Yes

Reviewer #2: Yes

2. Has the statistical analysis been performed appropriately and rigorously? 

Reviewer #1: I Don't Know

Reviewer #2: Yes

3. Have the authors made all data underlying the findings in their manuscript fully available?

Reviewer #1: Yes

Reviewer #2: Yes

4. Is the manuscript presented in an intelligible fashion and written in standard English?

Reviewer #1: Yes

Reviewer #2: Yes

5. Review Comments to the Author

Reviewer #1: This is an interesting study looking into the assessing COVID Severity combining viral genomics and clinical data.

I think methods section should come after the introduction. Figure and table legends should be separate at the end of the manuscript.

I think it's better to report any association as non-significant if p-value is indicating so and try to not use the term almost significant.

Please provide the reference for the following sentence "phacoemulsification-surgery to treat cataracts in the eye is

427 likely to have little bearing on severe COVID-19 pneumonia".

I could not find a clear conclusion section. Please consider adding a separate section in the abstract and the body of the manuscript.

Reviewer #2: The authors present an interesting topic regarding developing a data resource for understanding the factors associated with COVID-19 severity. The study combines viral genomics with clinical data to determine these factors. Please find my comments and suggestions below:

Although the topic of the study is very interesting, and the authors tried to combine viral genomics with the clinical data to provide a comprehensive source of understanding the risk factors of COVID-19 severity, the construction and flow of the manuscript could be clearer. For example, having the materials and methods section before the results will help the reader understand the design of the study before jumping to the results and conclusions.

The authors didn’t discuss the limitations that they may have faced in conducting this study; for example, the majority (75%) of all the cases were of white ethnic background, which will affect the generalizability of the study results.

Finally, I applaud the authors’ efforts for their attempt to understand the factors associated with COVID-19 severity, and I believe combining viral genomics with clinical data can provide a better overlook of the COVID-19 infection and its severity.

6. PLOS authors have the option to publish the peer review history of their article (what does this mean?). If published, this will include your full peer review and any attached files.

Reviewer #1: No

Reviewer #2: No

---

## [Author Response · Author response to Decision Letter 0]

7 Mar 2023

Journal Requirements

We have followed the provided style templates and made the following changes:

1. Updated Figures to be labelled as Fig 1, Fig, 2 etc throughout, and renamed corresponding files. 

2. Updated supplementary information and corresponding files to meet the requirements.

3. Ensured that all headings meet the requirements in terms of size, face and use of sentence case.

4. Added titles for all figure and table legends.

5. Updated Table names throughput to match the requirements.

6. Removed citation of other figures and tables from legends.

7. Updated citations to use square brackets in the text of the manuscript.

8. Relabelled supplementary figures to ensure labelling is sequential following movement of the Materials and methods section. 

9. Reformatted Tables so that the name and title are above the table, and the legend is below the table.

10. Used the “^” symbol for the group affiliation.

11. Removed the short subtitle of the manuscript (“Assessment of risk factors for severe COVID-19 in a large acute UK NHS hospital Trust”).

12. Removed postcode and added county information for affiliations as indicated in the style document.

13. Updated corresponding author information to match the style guide.

"The authors would like to thank all of the NHS patients and staff who supported this work to be undertaken throughout the pandemic. We also thank the COG-UK Consortium and its members for their constant support and assistance, in particular in funding this project under their Internal Principal Investigator Research Funding Scheme. We thank the UK National Institute for Health Research Clinical Research Network (NIHR CRN) for their support of the work."

"This work was primarily funded by the COVID-19 Genomics UK (COG-UK) consortium (https://www.cogconsortium.uk/), under their Internal Principal Investigator Research Funding Scheme. COG-UK is supported by funding from the Medical Research Council (MRC; https://www.ukri.org/councils/mrc/) part of UK Research & Innovation (UKRI; https://www.ukri.org/), the National Institute of Health Research (NIHR; https://www.nihr.ac.uk/) [grant code: MC_PC_19027], and Genome Research Limited, operating as the Wellcome Sanger Institute (https://www.sanger.ac.uk/). The authors acknowledge the use of data generated through the COVID-19 Genomics Programme funded by the Department of Health and Social Care (DHSC; https://www.gov.uk/government/organisations/department-of-health-and-social-care). The views expressed are those of the author and not necessarily those of the Department of Health and Social Care or PHE or UKHSA. MFM, AL, and OG were also supported by a UKRI Science & Technology Facilities Council (STFC) Impact Accelerator Account awarded to the Institute of Cosmology and Gravitation (ICG) at the University of Portsmouth. Additional funding for the project came from the University of Portsmouth Faculty of Science and Health (https://www.port.ac.uk/about-us/structure-and-governance/organisational-structure/our-academic-structure/faculty-of-science-and-health), and the Wessex Academic Health Sciences Centre (AHSC; https://wessexahsn.org.uk/). In addition, SCR and AHB are funded by Research England’s Expanding Excellence in England (E3) Fund. The primary funders had no role in study design, data collection and analysis, decision to publish, or manuscript preparation."

We have removed the part of the sentence in the Acknowledgement Section that reads “in particular in funding this project under their Internal Principal Investigator Research Funding Scheme” to ensure that the Acknowledgement Section does not contain information regarding funding. This funding is also already declared in the Funding statement in the first line, so does not need to be added to the current Funding Statement. 

We are not sure that the NIHR CRN support mentioned in this section would belong in the Funding Statement, as they supported the project not through funding, but through access to research infrastructure at the NHS hospital through adding our project to their research portfolio to support, for instance, sample and data collection (https://www.nihr.ac.uk/researchers/collaborations-services-and-support-for-your-research/run-your-study/crn-portfolio.htm). 

We therefore believe that the funding statement does not need to be modified, but please do let me know if this is incorrect.

We have removed the funding statement from the manuscript.

"Scott Elliot and Salman Goudarzi currently work for QIAGEN, UK. The remaining authors declare that no competing interests exist."

We note that one or more of the authors are employed by a commercial company: QIAGEN, UK.

Whilst these two authors now work for the commercial company QIAGEN, at the time that this work was conducted they were employed by Portsmouth Hospitals University NHS Trust (Scott Elliott) and the University of Portsmouth (Salman Goudarzi). We have not included QIAGEN amongst their affiliations in the title page, as their current positions were not relevant for the work as submitted. Therefore, I do not believe that the Funding Statement needs to be changed, as QIAGEN provided no support to the project, not even through salaries. 

Please update our Competing Interests statement to read as follows:

“Scott Elliot and Salman Goudarzi currently work for QIAGEN, UK, however were employees of Portsmouth Hospitals University NHS Trust and the University of Portsmouth respectively when the work described in this manuscript was conducted. QIAGEN had no role in the study design, data collection and analysis, decision to publish, or manuscript preparation, and this does not alter our adherence to PLOS ONE policies on sharing data and materials. The remaining authors declare that no competing interests exist.”

This statement has now been removed from the manuscript.

Please find the updated Competing Interests Statement above. However, as described above, we do not believe that any changes are necessary for the Funding Statement.

4. We noted in your submission details that a portion of your manuscript may have been presented or published elsewhere. "The whole genome sequencing data of SARS-CoV-2 samples used in this manuscript have been submitted to the COVID-19 Genomics UK Consortium database, and used to underpin a wide range of publications across the Consortium that have used the entire database to explore cases from across the UK. In addition, we have submitted a manuscript using these sequencing data that is currently undergoing review with Frontiers in Cellular and Infection Microbiology: Molecular Viral Pathogenesis. That manuscript focusses on using these data to identify shared infections and nosocomial transmission chains within the hospital amongst staff and patients, whilst the submitted manuscript combines viral genomic data with a range of other clinical data to assess risk factors for disease severity. These are thus very different studies, using the same dataset to address distinct but important questions, and thus do not constitute dual publication." Please clarify whether this publication was peer-reviewed and formally published. If this work was previously peer-reviewed and published, in the cover letter please provide the reason that this work does not constitute dual publication and should be included in the current manuscript.

The manuscript described in our original cover letter has now been published in Frontiers in Cellular and Infection Microbiology, and can be found here:

https://www.frontiersin.org/articles/10.3389/fcimb.2022.1066390/full#h10

In addition, the viral genomic data used throughout the manuscript are shared across the COG-UK consortium to create a large-scale resource for SARS-CoV-2 genomic data that has underpinned a very wide range of publications. However, these genomic data represent only a small element of the data used within our manuscript. The primary focus of the paper mentioned above was focussed around using information on patient and staff ward locations to track nosocomial spread, and explore the role played by the introduction of the vaccine on reducing spread. Importantly as well, the Frontiers paper covers only a narrow time range, from September 2020 to May 2021, whilst our submitted manuscript covers all cases from March 2020. 

In comparison, in our current manuscript we combined these genomic data with a unique data set of patient outcomes collected by Portsmouth Hospitals University NHS Trust, which provided a large range of data points for comparison with the genomic data. Whilst a brief comparison of patient outcomes between nosocomial groups was described in the Frontiers paper, the comparison of outcomes with patient demographics, role of Alpha variant cases on outcomes, identification of variants associated with increased disease severity, etc described in this paper are exploring a completely separate series of questions. 

We therefore strongly feel that, whilst ostensibly using some of the same data (primarily the genomic data), this work does not constitute dual publication and should be included in the current manuscript

As stated in our current Data Availability statement, the primary data set used for this study contains data which cannot be shared publicly due to containing potentially identifiable patient data. Whilst every effort was made to anonymise these data, given that all data are from patients from a single NHS Trust, the chances of these data representing a potential breach of patient confidentiality means that it cannot be widely shared. Indeed, our agreed ethical approval is based around not allowing these data to be made publicly available.

However, access to these data can be made available where necessary, through the Portsmouth Hospitals University NHS Trust Institutional Data Access/Ethics Committee, who will be able to assess whether sharing of the primary data is both ethical and appropriate. All genomic data from this study is shared publicly through both the European Nucleotide Archive (ENA) and GISAID database. These points have been specified in our current Data Availability statement, but please let us know if this is not suitable.

6. One of the noted authors is a group or consortium The COVID-19 Genomics UK (COG-UK) consortium. In addition to naming the author group, please list the individual authors and affiliations within this group in the acknowledgments section of your manuscript. Please also indicate clearly a lead author for this group along with a contact email address.

The COG-UK Consortium is a very large consortium, consisting of hundreds of members. The method for acknowledging all consortium members expected by the consortium is to share the current list of affiliations through a supplementary file, as we have done through S1 File. This group authorship is used to acknowledge the fact that most studies from COG-UK utilise data from the COG-UK CLIMB database, which consists of data generated by groups across the UK. It would therefore be impossible to unpick exactly who was responsible for each individual SARS-CoV-2 sequence, and thus the decision was made to include all COG-UK affiliates in aggregate. This is the standard operating procedure for COG-UK publications, and information on this can be found here (https://www.cogconsortium.uk/priority-areas/research/cog-uk-publications/). The style of referencing for the consortium that we have used has been used in a wide range of publications, including PLOS One (for instance, https://journals.plos.org/plosone/article?id=10.1371/journal.pone.0243185). We therefore do not believe that any changes are necessary. 

7. Your ethics statement should only appear in the Methods section of your manuscript. If your ethics statement is written in any section besides the Methods, please move it to the Methods section and delete it from any other section. Please ensure that your ethics statement is included in your manuscript, as the ethics statement entered into the online submission form will not be published alongside your manuscript. 

We have moved the Ethics Statement to the Materials and Methods section as requested. 

We have updated the reference list to ensure that the order of the references fits with the new order in the text, after movement of the Materials and methods section. In addition, we have added two additional references:

1) Cook et al. [56], which is the fully published citation for our previous study, which we previously cited in our manuscript as “Cook et al, under revision”.

2) Stuart et al. [47], which was added in response to one of the comments from Reviewer 1.

Otherwise, to the best of our knowledge, all other references are accurate. 

Reviewer 1 comments

This is an interesting study looking into the assessing COVID Severity combining viral genomics and clinical data.

We thank the reviewer for the time that they have spent in reviewing our manuscript. We are glad that they found it interesting, and have addressed their comments and concerns below.

I think methods section should come after the introduction. 

We have now moved the Materials and methods section to come after the Introduction as suggested. All references have been updated accordingly.

Figure and table legends should be separate at the end of the manuscript.

We thank the reviewer for this suggestion, however here we have followed the PLOS One submission guidelines, which state that “Figure captions must be inserted in the text of the manuscript, immediately following the paragraph in which the figure is first cited” and that tables should be placed “in your manuscript file directly after the paragraph in which it is first cited”. Full information can be found here (https://journals.plos.org/plosone/s/submission-guidelines#loc-figures-and-tables). 

I think it's better to report any association as non-significant if p-value is indicating so and try to not use the term almost significant.

Thank you for noting this issue, we agree that this is a term that should not have been used. It was previously corrected in the abstract prior to submission, but clearly missed in the Results section. We have adjusted the wording to ensure that this is highlighted as a non-significant change (line 581). 

Please provide the reference for the following sentence "phacoemulsification-surgery to treat cataracts in the eye is likely to have little bearing on severe COVID-19 pneumonia".

This statement was intended to simply be an example of a condition that, whilst encoded as a comorbidity, would be unlikely to directly influence COVID-19 severity. This was included to make the point that the number of comorbidities itself may be skewed by illnesses unlikely to be linked to pulmonary disease severity, but perhaps correlated to other comorbidities and demographic factors that are (such as age). 

We have modified the text in the manuscript to make this point clearer, and have added a reference to a paper from 2022 (Stuart et al. [47]), which looked at the incidence of risk factors for severe COVID-19 amongst cataract patients (lines 795 - 800). We hope that this has clarified this point.

I could not find a clear conclusion section. Please consider adding a separate section in the abstract and the body of the manuscript.

We thank the reviewer for this suggestion, and have added a specific Conclusion section to the main body of the manuscript (lines 1081-1102). The overall conclusion is that whilst certain factors relating to the health and demographics of the patient were associated with disease severity, the viral mutations were not. Thus, it is likely that further understanding of the factors influencing outcomes to COVID-19 will come from the study of host-specific effects, rather than virus-specific effects. We felt that this was already expressed in the final points of our abstract, so have not updated this further.

Reviewer #2: 

The authors present an interesting topic regarding developing a data resource for understanding the factors associated with COVID-19 severity. The study combines viral genomics with clinical data to determine these factors. Please find my comments and suggestions below:

We thank the reviewer for assessing our manuscript and providing comments and suggestions. We are glad that they found the topic interesting, and have answered the specific points below. We hope that these responses address any concerns. 

Although the topic of the study is very interesting, and the authors tried to combine viral genomics with the clinical data to provide a comprehensive source of understanding the risk factors of COVID-19 severity, the construction and flow of the manuscript could be clearer. For example, having the materials and methods section before the results will help the reader understand the design of the study before jumping to the results and conclusions.

We have now moved the Materials and methods section to come after the Introduction as suggested. All references have been updated accordingly.

The authors didn’t discuss the limitations that they may have faced in conducting this study; for example, the majority (75%) of all the cases were of white ethnic background, which will affect the generalizability of the study results.

We thank the reviewer for identifying this as an area missing from our discussion. We have added the following to our discussion, alongside discussing the low sample number, limitation of looking at only a single hospital site, and the fact that our study cannot draw conclusions for Delta nor Omicron variants (lines 1044-1047):

“One other key limitation of this study is that the demographics of the patient cohort are skewed for those of the local area, in particular with over 75% of those in the study being of a white background (Table 1). These results may therefore not be generalisable to the population as a whole. ”

Finally, I applaud the authors’ efforts for their attempt to understand the factors associated with COVID-19 severity, and I believe combining viral genomics with clinical data can provide a better overlook of the COVID-19 infection and its severity.

We thank the reviewer once again for their suggestions and kind words regarding our manuscript.

---

## [Editor Report · Decision Letter 1]

8 Mar 2023

Combining viral genomics and clinical data to assess risk factors for severe COVID-19 (mortality, ICU admission, or intubation) amongst hospital patients in a large acute UK NHS hospital Trust

PONE-D-22-31171R1

Dear Dr. Robson,

We’re pleased to inform you that your manuscript has been judged scientifically suitable for publication and will be formally accepted for publication once it meets all outstanding technical requirements.

Kind regards,

Divyansh Agarwal

Academic Editor

PLOS ONE
---

## [Editor Report · Acceptance letter]

14 Mar 2023

PONE-D-22-31171R1 

Combining viral genomics and clinical data to assess risk factors for severe COVID-19 (mortality, ICU admission, or intubation) amongst hospital patients in a large acute UK NHS hospital Trust 

Dear Dr. Robson:

I'm pleased to inform you that your manuscript has been deemed suitable for publication in PLOS ONE. Congratulations! Your manuscript is now with our production department. 

Kind regards, 

on behalf of

Dr. Divyansh Agarwal 

Academic Editor

PLOS ONE